# Learning Debiased Representation via Disentangled Feature Augmentation

**Jungsoo Lee**[*1,2]  **Eungyeup Kim**[*1,2]  **Juyoung Lee**[2]  **Jihyeon Lee**[1]  **Jaegul Choo**[1]

[1]KAIST AI, [2]Kakao Enterprise, South Korea

[1]{bebeto, eykim94, gina3833, jchoo}@kaist.ac.kr,
[2]{bebeto.lee, josh.ey, michael.jy}@kakaoenterprise.com

## Abstract

Image classification models tend to make decisions based on peripheral attributes of data items that have strong correlation with a target variable (*i.e.*, *dataset bias*). These *biased* models suffer from the poor generalization capability when evaluated on unbiased datasets. Existing approaches for debiasing often identify and emphasize those samples with no such correlation (*i.e.*, *bias-conflicting*) without defining the bias type in advance. However, such bias-conflicting samples are significantly scarce in biased datasets, limiting the debiasing capability of these approaches. This paper first presents an empirical analysis revealing that training with "diverse" bias-conflicting samples beyond a given training set is crucial for debiasing as well as the generalization capability. Based on this observation, we propose a novel feature-level data augmentation technique in order to synthesize diverse bias-conflicting samples. To this end, our method learns the disentangled representation of (1) the *intrinsic attributes* (*i.e.*, those inherently defining a certain class) and (2) *bias attributes* (*i.e.*, peripheral attributes causing the bias), from a large number of *bias-aligned* samples, the bias attributes of which have strong correlation with the target variable. Using the disentangled representation, we synthesize bias-conflicting samples that contain the diverse intrinsic attributes of bias-aligned samples by swapping their latent features. By utilizing these diversified bias-conflicting features during the training, our approach achieves superior classification accuracy and debiasing results against the existing baselines on synthetic and real-world datasets.

## 1 Introduction

Despite the recent advancement of deep neural networks, they often rely overly on the correlation between peripheral attributes and labels, referred to as *dataset bias* [1], especially when such strong bias is found in a given dataset. A majority of samples in the biased dataset exhibit visual attributes that are not innate but frequently co-occur with target labels (*i.e.*, *bias attributes*). For example, most of the bird images in the training dataset may contain the background as the blue sky, while the birds may still be found in different places. Thus, the model trained with such a biased dataset is likely to learn the bias attributes more than *intrinsic attributes*, the innate visual attributes that inherently define a certain class, *e.g.*, the wings of birds. This causes the model to learn shortcuts for classification [2], failing to generalize on the images with no such correlations (*e.g.*, birds on grounds or grass) during the test phase. Throughout the paper, *bias-aligned* samples correspond to data items containing a strong correlation between bias attributes and labels (*e.g.*, birds in the sky), while *bias-conflicting* samples indicate the other cases that are rarely found (*e.g.*, birds on grounds).

To tackle such a task, previous studies often define a specific bias type (*e.g.*, color and texture) in advance [3, 4, 5, 6, 7, 8, 9, 10], which enables them to design a debiasing network tailored for the predefined bias type. For example, Bahng *et al.* [6] leverage BagNet [11], which has limited size

---

* indicates equal contribution. The order of first authors was chosen by tossing a coin.

35th Conference on Neural Information Processing Systems (NeurIPS 2021).

of receptive fields, to focus on learning color and texture. However, defining a bias type in advance 1) limits the capability of debiasing in other bias types and 2) requires expensive labor to manually identify the bias type. To handle such an issue, a recent approach [12] defines a bias based on an intuitive observation that the bias attributes are often *easier* to learn than the intrinsic attributes for neural networks. In this regard, they re-weight bias-conflicting samples while de-emphasizing the bias-aligned ones. However, we point out that the reason behind the limited generalization capability of existing debiasing approaches lies in the significant scarcity of bias-conflicting samples compared to the bias-aligned ones in a given training set. In other words, it is challenging to learn the debiased representation from these scarce bias-conflicting samples because the models are prone to memorize (thus being overfitted to) these samples, failing to learn the intrinsic attributes. Therefore, we claim that a neural network can learn properly debiased representation when these data items are diversified during training.

We conduct a brief experiment to demonstrate the importance of *diversity* in debiasing. Diversity in our work indicates the different valid realization of intrinsic attributes in a certain class (*e.g.*, thick, narrow, tilted, and scribbled digit shapes in MNIST [13]). Our observation is that training a model with diverse bias-conflicting samples beyond a given training set is crucial for learning debiased representation (Section 3.2). In this regard, synthesizing bias-conflicting samples is one of the straightforward approaches to increase the diversity of such samples. In fact, a large amount of bias-aligned samples in a given training set already contain diverse intrinsic attributes, which can work as informative sources for increasing the diversity. However, as bias and intrinsic attributes are highly entangled in their embedding space, it is difficult to extract the intrinsic ones from these bias-aligned samples. Therefore, disentangling these correlations enables to synthesize diversified bias-conflicting samples that originate from bias-aligned samples.

In this paper, we propose a novel feature augmentation approach via disentangled representation for debiasing. We first train two different encoders to embed images into the disentangled representation of their intrinsic and bias attributes. With the disentangled representation, we randomly swap the latent vectors extracted from different images, most of which are bias-aligned samples in our training set. These swapped features thus contain both bias and intrinsic attributes without the correlation between them, which, in turn, can work as augmented bias-conflicting samples in our training. These features include intrinsic features of bias-aligned ones, increasing the diversity of a given training set, especially for bias-conflicting data items. Furthermore, to enhance the quality of diversified features, we propose a scheduling strategy of feature augmentation which enables to utilize the representation disentangled to a certain degree. In summary, the main contributions of our work include:

- Through our preliminary experiment, we reveal that increasing the diversity of bias-conflicting samples is crucial for debiasing.

- Based on such an observation, we propose a novel feature augmentation method via disentangled representation for diversifying the bias-conflicting samples.

- We achieve the state-of-the-art performances in two synthetic datasets (*i.e.*, Colored MNIST and Corrupted CIFAR-10) and one real-world dataset (*i.e.*, Biased FFHQ) against existing baselines.

## 2 Related Work

**Debiasing predefined bias** Several existing approaches mitigate the bias by pre-defining a certain bias type, either explicitly [3, 4, 5] or implicitly [6, 7, 8, 9, 10, 14]. For example, Bahng *et al.* [6] and Wang *et al.* [7] design a color- and texture-oriented network to adversarially learn a debiased model against the biased one. However, as these methods still require a specific bias type such as texture in advance, they lack the general applicability to the datasets where the bias types are demanding to recognize.

Instead of defining certain types of bias, recent approaches [12, 15, 16] rely on the straightforward assumption that networks are prone to exploit the bias when it acts as a shortcut [2], *i.e.*, easy to learn in the early training phase. Nam *et al.* [12] emphasize the bias-conflicting samples during training by using generalized cross-entropy loss [17]. Darlow *et al.* [15] and Huang *et al.* [16] presume that high gradient of latent vectors accounts for the shortcuts that model learns. In the line with the recent studies, we tackle debiasing without pre-defining a certain bias type.

| Dataset | Diversity ratio | Sampling ratio | Accuracy (%) |
|---------|-----------------|----------------|--------------|
| | 5% | 50% | **83.77**$_{\pm 2.03}$ |
| Colored MNIST | 1% | 50% | 67.19$_{\pm 1.99}$ |
| | 5% | 1% | 77.97$_{\pm 6.00}$ |
| | 1% | 1% | 49.91$_{\pm 4.22}$ |
| | 5% | 50% | **46.99**$_{\pm 0.82}$ |
| Corrupted CIFAR-10 | 1% | 50% | 33.08$_{\pm 0.80}$ |
| | 5% | 1% | 36.66$_{\pm 0.55}$ |
| | 1% | 1% | 23.98$_{\pm 0.00}$ |

Table 1: The classification accuracy on the unbiased test sets. The diversity ratio indicates the ratio of bias-conflicting samples in the dataset pooled for each experiment. The sampling ratio refers to the ratio of bias-conflicting samples included in each mini-batch. We report the averaged accuracy over three independent trials with the standard deviation. In both datasets, we observe that the bias can be mitigated with diverse bias-conflicting samples even with a small sampling ratio. Bold and underlined values indicate the best and second best accuracy, respectively.

**Data augmentation for debiasing** Geirhos *et al.* [10] mitigate the texture bias by utilizing additional training images with their styles being transferred by adaptive instance normalization (AdaIN) [18]. Minderer *et al.* [19] train an image-to-image translation network for removing shortcut cues in the self-supervised task. However, such image-level data augmentation is limited to resolving the predefined texture bias which can not be adopted to other general types of bias.

One alternative is to exploit the latent space for data augmentation. For example, Darlow *et al.* [15] adversarially perturb the latent vectors corresponding to the high gradients to generate the samples against bias. Zhou *et al.* [20] mix the style of different source domains by AdaIN [18] to increase the domain generalization ability. Despite the effectiveness of the augmentation in the latent space, the strong unwanted correlation between bias attributes and labels prevents from obtaining the desirable intrinsic features. We resolve this issue by leveraging the disentangled representation in debiasing, which is widely used in image-to-image translation task [21, 22, 23]. To the best of our knowledge, no previous work in debiasing leverage this disentangled representation for the purpose of feature augmentation. For the rest of the paper, we elaborate how we perform the feature augmentation based on the disentangled representation.

## 3 Importance of Diversity in Debiasing

This section describes the details of a toy-set experiment in which we observe the importance of diversity in learning debiased representation. In Section 3.1, we first introduce the two synthetic datasets, Colored MNIST and Corrupted CIFAR-10, that we utilize for the observation. Then, we elaborate the results of the experiments in Section 3.2.

### 3.1 Dataset

**Colored MNIST** is a modified MNIST dataset [13] with the color bias. We select ten distinct colors and inject each color on the foreground of each digit to create color bias. By adjusting the number of bias-conflicting data samples in the training set, we obtain four different datasets with the ratio of bias-conflicting samples of 0.5%, 1%, 2%, and 5%.

**Corrupted CIFAR-10** has ten different types of texture bias applied in CIFAR-10 [24] dataset, constructed by following the design protocol of Hendrycks and Dietterich [25]. Each class is highly correlated with a certain texture (*e.g.*, frost and brightness). Corrupted CIFAR-10 also has four different datasets with their correlation ratios as in Colored MNIST.

### 3.2 Increasing diversity outperforms oversampling

To confirm the significance of diversity of bias-conflicting samples in debiasing, we train four different settings: oversampling bias-conflicting samples by 50% in each mini-batch (*i.e.*, 128 from a batch size of 256), from the pool of i) 5% dataset and ii) 1% dataset, sampling bias-conflicting samples by 1% in each mini-batch (*i.e.*, 2 from a batch size of 256) from the pool of iii) 5% dataset and iv) 1% dataset. Oversampling provides the same amount of bias-conflicting samples as the aligned ones to

the model in every training step. Bias-conflicting images sampled from the pool of 5% dataset have more diverse appearances of bias-conflicting samples compared to those from 1% dataset.

Table 1 shows the image classification accuracy of each setting validated on the unbiased test images. Apparently, oversampling diverse bias-conflicting samples (first row) outperforms the other three methods. Similarly, sampling a small amount of bias-conflicting samples with the least diversity (fourth row) shows the lowest classification accuracy. The interesting finding is that sampling *fewer but diverse* conflicting samples in each mini-batch (third row) outperforms oversampling bias-conflicting samples with limited diversity (second row). These results lead to the conclusion that the diversity of bias-conflicting samples is a more crucial factor for learning debiased representation than the ratio of sampling in the training. As the diversity is limited (the latter case), the model can be easily overfitted to the given bias-conflicting samples, thus less likely to learn the generalized intrinsic attributes. With the Colored MNIST as an example, the shape of digits may vary. To be more specific, the digit shape may be thick, narrow, tilted, scribbled, and etc. If the bias-conflicting samples do not include certain visual facets (e.g., not including scribbled digit images) due to the limited number of samples, the model may imperfectly learn the intrinsic attributes of digit shapes. On the other hand, in the former case (third row), the model can learn multiple facets of intrinsic attributes when they are sampled from the diverse pool of datasets, resulting in learning intrinsic attributes even without oversampling the bias-conflicting images.

## 4 Debiasing via disentangled feature augmentation

Motivated by such an observation in Section 3.2, we propose a feature-level augmentation strategy for synthesizing additional bias-conflicting samples, as illustrated in Fig. 1. First, we train the two separate encoders which embed an image into disentangled latent vectors corresponding to the intrinsic and bias attributes, respectively (Section 4.1). Swapping these feature vectors among training samples enables to augment the bias-conflicting samples which no more contain a correlation between two attributes (Section 4.2). To further enhance the effectiveness, we schedule the feature augmentation after the representation is disentangled at a certain degree (Section 4.3).

### 4.1 Learning disentangled representation

In contrast to the bias-conflicting samples, a large amount of bias-aligned images have diverse appearances of their intrinsic attributes. By leveraging these attributes for augmentation, we can naturally obtain the diversified bias-conflicting samples containing the diverse intrinsic attributes. However, it remains challenging in that these attributes are strongly correlated with the bias attributes in the bias-aligned samples. Therefore, we propose to design two encoders with their linear classifiers to extract the disentangled latent vectors from the input images.

As shown in Fig. 1, encoders $E_i$ and $E_b$ embed an image $x$ into intrinsic feature vectors $z_i = E_i(x)$ and bias feature vectors $z_b = E_b(x)$, respectively. Afterward, linear classifiers $C_i$ and $C_b$ take the concatenated vector $z = [z_i; z_b]$ as input to predict the target label $y$. To train $E_i$ and $C_i$ as intrinsic feature extractor and $E_b$ and $C_b$ as bias extractor, we utilize the relative difficulty score of each data sample, proposed in the previous work of Nam *et al.* [12]. More specifically, we train $E_b$ and $C_b$ to be overfitted to the bias attributes by utilizing the generalized cross entropy (GCE) [17], while $E_i$ and $C_i$ are trained with the cross entropy (CE) loss. Then, the samples with high CE loss from $C_b$ can be regarded as the bias-conflicting samples compared to the samples with low CE loss. In this regard, we obtain the relative difficulty score of each data sample as

$$W(z) = \frac{CE(C_b(z), y)}{CE(C_i(z), y) + CE(C_b(z), y)}. \tag{1}$$

As bias-conflicting samples obtain high values of $W$, we emphasize the loss of these samples for training $E_i$ and $C_i$, enforcing them to learn the intrinsic attributes. Therefore, the objective function for disentanglement can be written as

$$L_{\text{dis}} = W(z)CE(C_i(z), y) + \lambda_{\text{dis}}GCE(C_b(z), y). \tag{2}$$

To ensure that $C_i$ and $C_b$ predicts target labels mainly based on $z_i$ and $z_b$, respectively, the loss from $C_i$ is not backpropagated to $E_b$, and vice versa.

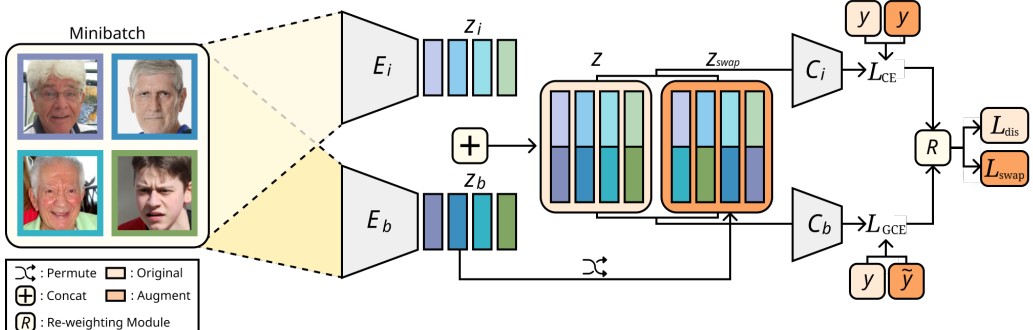

Figure 1: The overview of our proposed debiasing approach. $(E_i, C_i)$ and $(E_b, C_b)$ are pairs of an encoder and a linear classifier trained to learn the disentangled representation of intrinsic attributes and bias attributes, respectively. With the disentangled features $z_i$ and $z_b$, the feature augmentation is performed by swapping these latent vectors among different training samples, after certain iterations of training. $R$ refers to the re-weighting algorithm which implicitly differentiates bias-aligned samples and bias-conflicting samples. Each color indicates the different data samples.

---

**Algorithm 1** Debiasing with disentangled feature augmentation

**Input:** image $x$, label $y$, iteration $t$, augment iteration $t_{\text{swap}}$
Initialize two networks $(E_i, C_i)$, $(E_b, C_b)$
**while** *not converged* **do**
    Extract $z_i$, $z_b$ from $E_i(x)$, $E_b(x)$
    Concatenate $z = [z_i; z_b]$
    Update $(E_i, C_i)$, $(E_b, C_b)$ with $L_{\text{dis}} = W(z)CE(C_i(z), y) + GCE(C_b(z), y)$
    if $t > t_{\text{swap}}$:
        Randomly permute $z = [z_i, z_b]$ into $z_{\text{swap}} = [z_i; \tilde{z}_b]$
        Calculate $L_{\text{swap}} = W(z)CE(C_i(z_{\text{swap}}), y) + GCE(C_b(z_{\text{swap}}), \tilde{y})$
        Update $(E_i, C_i)$, $(E_b, C_b)$ with $L_{\text{total}} = L_{\text{dis}} + \lambda_{\text{swap}} L_{\text{swap}}$
**end**

---

## 4.2 Feature swapping for augmentation

While such an architecture disentangles the intrinsic features and bias features, $E_i$ and $C_i$ are still mainly trained with an excessively small amount of bias-conflicting samples. Therefore, $E_i$ and $C_i$ fail to fully acquire the intrinsic representation of a target class. To promote further improvement in learning intrinsic feature vectors, we diversify the bias-conflicting samples by swapping the disentangled latent vectors among the training sets. In other words, we randomly permute the intrinsic features and bias features in each mini-batch and obtain $z_{\text{swap}} = [z_i; \tilde{z}_b]$ where $\tilde{z}_b$ denotes the randomly permuted bias attributes of $z_b$. As the intrinsic and bias attributes in $z_{\text{swap}}$ are obtained from two different images, they certainly have less correlation compared to $z = [z_i; z_b]$ where both are from the same image. Since the biased dataset is mostly composed of bias-aligned samples, these vectors are likely from the bias-aligned samples, highly diversified compared to the bias-conflicting ones. Then, $z_{\text{swap}} = [z_i; \tilde{z}_b]$ act as augmented bias-conflicting latent vectors with diversity inherited from the bias-aligned samples. Along with $L_{\text{dis}}$, we add the following loss function to train two neural networks with the augmented features

$$L_{\text{swap}} = W(z)CE(C_i(z_{\text{swap}}), y) + \lambda_{\text{swap}_b} GCE(C_b(z_{\text{swap}}), \tilde{y}), \tag{3}$$

where $\tilde{y}$ denotes target labels for permute bias attributes $\tilde{z}$. Thus, total loss function is described as

$$L_{\text{total}} = L_{\text{dis}} + \lambda_{\text{swap}} L_{\text{swap}} \tag{4}$$

where $\lambda_{\text{swap}}$ is adjusted for weighting the importance of the feature augmentation.

## 4.3 Scheduling the feature augmentation

While training with additional synthesized features helps to mitigate the unwanted correlation, utilizing them from the beginning of training does not improve the debiasing performance. To be more

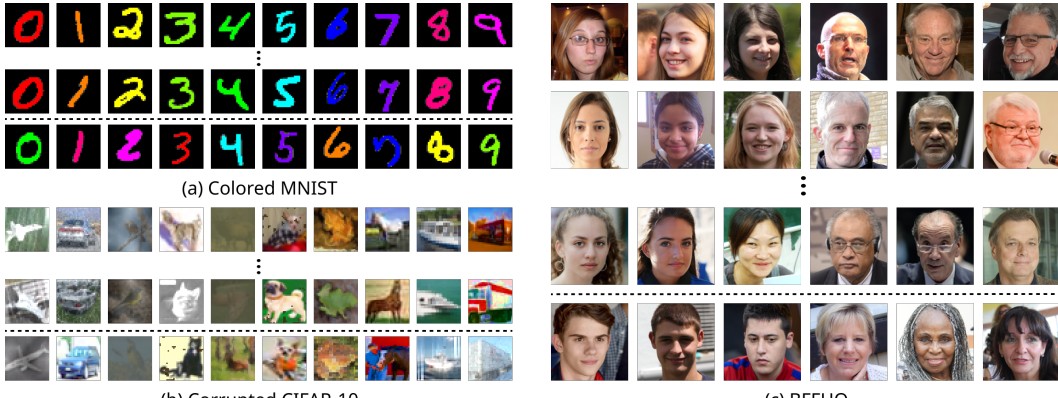

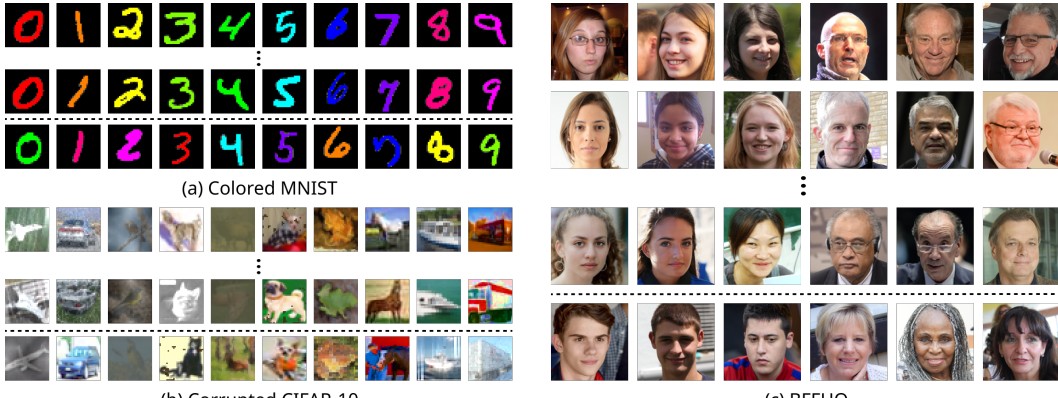

(a) Colored MNIST

(b) Corrupted CIFAR-10

(c) BFFHQ

Figure 2: Example images of datasets utilized in our work. In each dataset, the images above the dotted line indicate the bias-aligned samples while the ones below the dotted line are the bias-conflicting samples. For Colored MNIST and Corrupted CIFAR-10, each column indicates each class. For BFFHQ, the group of three columns indicates each class.

specific, in the early stage of training, the representations of $z_i$ and $z_b$ are imperfectly disentangled to be used as the sources of feature augmentation. Feature augmentation should be conducted after two features are disentangled at a certain degree. Without the disentangled representation, the augmented features work as noisy samples which may aggravate the debiasing performances. We verify the importance of scheduling the feature augmentation in Table 3. Our approach can be summarized with Algorithm 1.

## 5 Experiment

This section demonstrates the effectiveness of feature augmentation based on disentangled representation in debiasing with both quantitative and qualitative evaluation. We compare our method with the previous approaches in debiasing with three different datasets with varied bias ratios. Then, we conduct the ablation study which demonstrates the importance of 1) learning disentangled representation, 2) feature augmentation, and 3) scheduling feature augmentation. For the qualitative evaluation, we verify how our approach disentangles the intrinsic features and bias features by visualizing them on 2D embedding space via t-SNE [26] and reconstructing images from them.

### 5.1 Experiment details

**Baselines** Our baselines consist of vanilla network, HEX [7], EnD [27], ReBias [6] and LfF [12]. Vanilla denotes the classification model trained only with the original cross-entropy (CE) loss, without any debiasing strategies. EnD explicitly leverages the bias labels (e.g., the color label in Colored MNIST) during the training phase. HEX and ReBias explicitly presume the texture of an image as a bias type, while LfF requires no prior knowledge on it.

**Datasets** As shown in Fig. 2, we use two synthetic datasets (Colored MNIST and Corrputed CIFAR-10) and one real-world dataset (Biased FFHQ) to evaluate the generalization of debiasing baselines over various domains. Biased FFHQ (BFFHQ) is curated from FFHQ dataset [28] which contains human face images annotated with their facial attributes. Among the facial attributes, we select age and gender as the intrinsic and bias attribute, respectively, and construct the dataset with images of high correlation between them. More specifically, most of the females are 'young' (*i.e.*, age ranging from 10 to 29) and males are 'old' (*i.e.*, age ranging from 40 to 59). Therefore, bias-aligned samples which compose the majority of the dataset are young women and old men.

For each dataset, we set the degree of correlation by adjusting the number of bias-conflicting samples among the training dataset. The ratio of bias-conflicting samples are 0.5%, 1%, 2% and 5% for both Colored MNIST and Corrupted CIFAR-10, respectively, and 0.5% for BFFHQ. For the evaluation of Colored MNIST and Corrupted CIFAR-10, we construct an *unbiased* test set which includes images without the high correlation existing in the training set. For the BFFHQ, we construct a *bias-conflicting* test set which excludes the bias-aligned samples from the *unbiased* test set. The reason is as following. The bias-aligned images consist a half of the unbiased test set in BFFHQ which

| Dataset | Ratio (%) | Vanilla [29] ✗ | HEX [7] ✓ | EnD [27] ✓ | ReBias [6] ✓ | LfF [12] ✗ | Ours ✗ |
|---|---|---|---|---|---|---|---|
| Colored MNIST | 0.5 | $35.19_{\pm3.49}$ | $30.33_{\pm0.76}$ | $34.28_{\pm1.20}$ | $\mathbf{70.47}_{\pm1.84}$ | $52.50_{\pm2.43}$ | $\underline{65.22}_{\pm4.41}$ |
| | 1.0 | $52.09_{\pm2.88}$ | $43.73_{\pm5.50}$ | $49.50_{\pm2.51}$ | $\mathbf{87.4}_{\pm0.78}$ | $61.89_{\pm4.97}$ | $\underline{81.73}_{\pm2.34}$ |
| | 2.0 | $65.86_{\pm3.59}$ | $56.85_{\pm2.58}$ | $68.45_{\pm2.16}$ | $\mathbf{92.91}_{\pm0.15}$ | $71.03_{\pm2.44}$ | $\underline{84.79}_{\pm0.95}$ |
| | 5.0 | $82.17_{\pm0.74}$ | $74.62_{\pm3.20}$ | $81.15_{\pm1.43}$ | $\mathbf{96.96}_{\pm0.04}$ | $80.57_{\pm3.84}$ | $\underline{89.66}_{\pm1.09}$ |
| Corrupted CIFAR-10 | 0.5 | $23.08_{\pm1.25}$ | $13.87_{\pm0.06}$ | $22.89_{\pm0.27}$ | $22.27_{\pm0.41}$ | $\underline{28.57}_{\pm1.30}$ | $\mathbf{29.95}_{\pm0.71}$ |
| | 1.0 | $25.82_{\pm0.33}$ | $14.81_{\pm0.42}$ | $25.46_{\pm0.41}$ | $25.72_{\pm0.20}$ | $\underline{33.07}_{\pm0.77}$ | $\mathbf{36.49}_{\pm1.79}$ |
| | 2.0 | $30.06_{\pm0.71}$ | $15.20_{\pm0.54}$ | $31.31_{\pm0.35}$ | $31.66_{\pm0.43}$ | $\underline{39.91}_{\pm0.30}$ | $\mathbf{41.78}_{\pm2.29}$ |
| | 5.0 | $39.42_{\pm0.64}$ | $16.04_{\pm0.63}$ | $40.26_{\pm0.85}$ | $43.43_{\pm0.41}$ | $\underline{50.27}_{\pm1.56}$ | $\mathbf{51.13}_{\pm1.28}$ |
| BFFHQ | 0.5 | $56.87_{\pm2.69}$ | $52.83_{\pm0.90}$ | $56.87_{\pm1.42}$ | $59.46_{\pm0.64}$ | $\underline{62.2}_{\pm1.0}$ | $\mathbf{63.87}_{\pm0.31}$ |

Table 2: Image classification accuracy evaluated on unbiased test sets of Colored MNIST and Corrupted CIFAR-10, and the bias-conflicting test set of BFFHQ with varying ratio of bias-conflicting samples. We denote whether the model requires a bias type in advance by *cross* mark (*i.e.*, not required), and *check* mark (*i.e.*, required). Best performing results are marked in bold, while second-best results are denoted with underlines.

may still be correctly classified by the biased classifier. This inflates the accuracy of the unbiased test set which is not our original intention. Therefore, we intentionally use the bias-conflicting test set for the BFFHQ.

**Implementation details** We use multi-layer perceptron (MLP) with three hidden layers for Colored MNIST, and ResNet-18 [29] for the remaining datasets. To accommodate the disentangled vectors, we double the number of hidden units in the last fully-connected layer of each network. During the inference phase, we use $C_i(z)$ for the final prediction, where $z = [z_i; z_b]$. For the training, we set the batch size of 256 for Colored MNIST and Corrupted CIFAR-10, respectively, and 64 for BFFHQ. Bias-conflicting augmentation is scheduled to be applied after 10K iterations for all datasets. We report the averaged accuracy of the unbiased test sets over three independent trials with the mean and the standard deviation. We include the remaining implementation details in Section D in the supplementary material.

## 5.2 Quantitative evaluation

**Comparison on test sets** Table 2 shows the comparisons of image classification accuracy evaluated on the test sets. In general, our approach demonstrates the superior performance in both synthetic and real-world datasets against the baselines with large gaps. Especially, compared to the baselines which do not define the bias types in advance (vanilla [29] and LfF [12]), our approach achieves the state-of-the-art performance across all datasets. This indicates that utilizing the diversified bias-conflicting samples through our augmentation plays a pivotal role in learning debiased representation regardless of the bias types.

Regarding the real-world dataset, our approach also outperforms HEX [7] and ReBias [6] which utilize the tailored modules for a specific bias type (*e.g.*, color and texture), and EnD [27] that uses the explicit bias labels. We even show superior performance compared to HEX in Colored MNIST without defining the bias type beforehand. While ReBias achieves the best accuracy in Colored MNIST, they utilize BagNet [11] in order to focus on the color bias. Even without using such an architecture, we achieve the second best performance which is comparable to ReBias.

**Ablation studies** Table 3 demonstrates the importance of each module in our approach through ablation studies: 1) disentangled representation learning, 2) feature augmentation, and 3) scheduling feature augmentation. We set the ratio of bias-conflicting samples to 1% for Colored MNIST and Corrupted CIFAR10, and 0.5% for BFFHQ. We also compare each module with the vanilla network (first row). We observe that performing the scheduled feature augmentation shows the best classification accuracy on the test sets across all datasets. We also show that performing feature augmentation at the early stage of training does not guarantee the effectiveness of debiasing. Performing feature augmentation at the beginning of training rather aggravates the performance. That is, when the representation of intrinsic attributes and bias attributes are not disentangled at a certain degree, augmented features may act as noisy samples. Training with these additional noisy features prevents models from achieving further improvement.

| Disentangle | Augment | Scheduled Augment | Colored MNIST | Corrupted CIFAR10 | BFFHQ |
|:---:|:---:|:---:|:---:|:---:|:---:|
| − | − | − | $52.09_{\pm 2.88}$ | $25.82_{\pm 0.33}$ | $56.87_{\pm 2.69}$ |
| ✓ | − | − | $74.03_{\pm 2.40}$ | $27.73_{\pm 1.02}$ | $59.4_{\pm 2.46}$ |
| ✓ | ✓ | − | $72.29_{\pm 3.82}$ | $32.81_{\pm 2.47}$ | $61.27_{\pm 3.26}$ |
| ✓ | ✓ | ✓ | $\mathbf{81.73}_{\pm 2.34}$ | $\mathbf{52.31}_{\pm 1.00}$ | $\mathbf{63.87}_{\pm 0.31}$ |

Table 3: Ablation studies on 1) disentangled representation learning, 2) feature augmentation, and 3) scheduling feature augmentation. Each row indicates the different training settings with *check* mark denoting the setting applied. We average the accuracy of each training over three independent trials.

## 5.3 Analysis

**2D Projection of Disentangled Representation** Fig. 3 shows the projection of latent vectors $z_i$ and $z_b$ extracted from the intrinsic encoder $E_i$ and bias encoder $E_b$, respectively, on a 2D space using Colored MNIST. We show projection of $z_i$ and $z_b$ in Fig. 3(a) and Fig. 3(b), respectively. The colors of projected dots in the first row (i) and the second row (ii) indicate the target labels and bias labels, respectively. We observe that $z_i$ are clustered according to the target labels while $z_b$ are clustered with the bias labels. The results represent that our method successfully learns the disentangled intrinsic and bias attributes.

**Prediction with Disentangled Representation** In Table 4, we report the 1) *original* and 2) *swapping* accuracy of $C_i$ and $C_b$, the linear classifiers of the intrinsic and the bias encoder, respectively. To be specific, for the *original* accuracy, we extract the two disentangled vectors, $z_i$ and $z_b$, from the same image, concatenate them to make $z = [z_i; z_b]$, and forward them into each linear classifier. For the *swapping* accuracy, however, we first permute $z_b$ and concatenate $z_i$ with the permuted $z_b$ (*i.e.*, denoted as $\tilde{z}_b$ in Section 4.2) to make $z_{\text{swap}} = [z_i; \tilde{z}_b]$. Then, we pass these concatenated latent vectors to each linear classifier. Afterward, we evaluate the accuracy of predicted labels of 1) $C_i(z)$ and $C_i(z_{\text{swap}})$ with intrinsic labels and 2) $C_b(z)$ and $C_b(z_{\text{swap}})$ with bias labels. The `Intrinsic` and `Bias` columns in Table 4 denote the accuracy with respect to the target labels and bias labels, respectively. Even the feature vectors of bias attributes are randomly swapped, our method maintains a reasonable classification accuracy. This indicates that our model well disentangles between $z_i$ and $z_b$, and $C_i$ robustly utilizes $z_i$ to predict target labels even when $z_b$ is taken from the different image, and vice versa. Note that we utilized the parameters of the model trained on each dataset after converging at a certain degree.

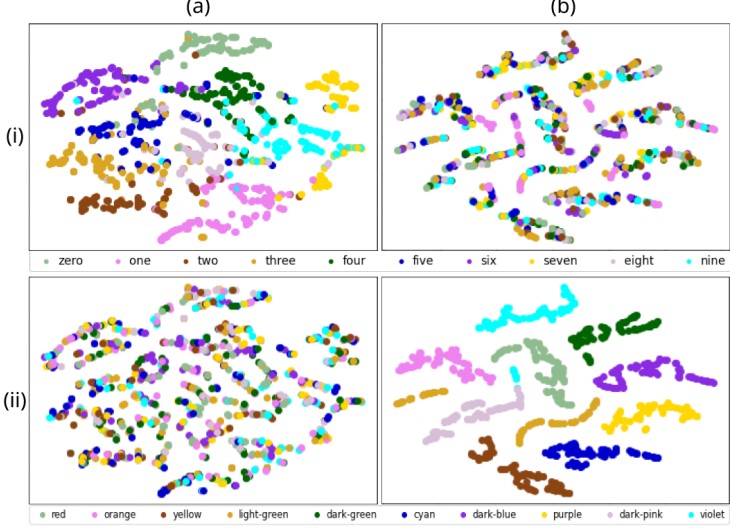

Figure 3: Each row (`i` and `ii`) include 2D projection of $z_i$ and $z_b$ with the colors encoded by their labels (*i.e.*, groundtruth labels in row `i` and bias labels in row `ii`) in Colored MNIST. We observe that $z_i$ and $z_b$ are well clustered according to the target and bias labels, respectively.

| Accuracy(%) | Colored MNIST | | Corrupted CIFAR10 | | BFFHQ | |
|---|---|---|---|---|---|---|
| | Intrinsic | Bias | Intrinsic | Bias | Intrinsic | Bias |
| Original | **76.08** | **98.07** | **35.63** | 74.16 | 57.40 | 49.00 |
| Swapping | 71.40 | 94.29 | 35.14 | **76.46** | **58.40** | **51.60** |

Table 4: Accuracy from disentangled representation. The ratio of bias-conflicting samples in Colored MNIST, Corrupted CIFAR-10, and BFFHQ are 1%, 1%, and 0.5%, respectively.

Figure 4: Reconstructed images from disentangled representation in Colored MNIST. Each column and row indicate the samples where the bias attribute (color) and the intrinsic attribute (digit) are extracted, respectively. By swapping the bias features with a given intrinsic feature, we observe that the color changes while maintaining the digit.

**Reconstruction of Disentangled Representation** Fig. 4 shows the reconstructed images of Colored MNIST by using the disentangled representation of intrinsic features and bias features. Images in the first row and column indicate the images used for extracting the bias attribute (*i.e.*, color) and intrinsic attribute (*i.e.*, digit), respectively. We train an auxiliary decoder by providing the latent vector $z$ from our pre-trained models as input in order to visualize the disentangled representations at the pixel level. By changing the bias attributes (as the column changes), the color of digit changes while maintaining the digit shape. This demonstrates that the bias features and intrinsic features independently contain color and digit information, respectively. Note that the reconstruction loss for updating the decoder is not backpropagated to our pre-trained classification models. Due to this fact, the reconstructed images may lack qualities such as showing blurry images. Further implementation details are included in Section D in the supplementary material.

## 6 Conclusions

In this work, we propose a feature augmentation method based on the disentangled representation of intrinsic and bias attributes. The main intuition behind our work is that increasing the diversity of bias-conflicting samples beyond a given training set is crucial for debiasing. Since the biased dataset strongly correlates the bias attributes and labels, we intentionally train two different encoders and extract bias features and intrinsic features. After the representations are disentangled to a certain degree, we proliferate the bias-conflicting samples by randomly swapping the vectors. We demonstrate the effectiveness of feature augmentation via extensive experiments, ablation studies, and qualitative evaluation of the disentangled representation. We believe our work inspires the future work of learning debiased representation with the improved generalization capability.

**Acknowledgements** This work was supported by the Institute of Information & communications Technology Planning & Evaluation (IITP) grant funded by the Korean government(MSIT) (No. 2019-0-00075, Artificial Intelligence Graduate School Program(KAIST), No. 2021-0-01778, Development of human image synthesis and discrimination technology below the perceptual threshold), the Air Force Research Laboratory, under agreement number FA9550-18-S-0003, and Kakao Enterprise. This material is based on research sponsored by The U.S. Government is authorized to reproduce and distribute reprints for Governmental purposes notwithstanding any copyright notation thereon.

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
