This supplementary material presents additional results and descriptions of our approach that are not included in the main paper due to the page limit. Section A shows the classification accuracy on Corrupted CIFAR-10 Type 0 and Type 1, the same datasets used in LfF [1]. Section B shows the reconstructed images of BFFHQ [2] by using the latent vectors of the intrinsic and the bias attributes. Section C explains how our method guarantees the disentangled representation between latent vectors of intrinsic and bias attributes. Afterwards, we illustrate the implementation details including architecture designs and hyper-parameters for training in Section D. Lastly, Section E briefly discusses broader impacts and limitations of our work in the related field.

## A    Corruption types in Corrupted CIFAR10

While we randomly sample the 10 corruption types among 20 types for constructing the Corrupted CIFAR10, Nam *et al.* [1] build two sets of Corrupted CIFAR10 with 10 corruption types each, which were termed as 'Type 0' and 'Type1'. In order to maintain the consistency of experimental setup with Nam *et al*, we also demonstrate the classification accuracy using the Corrupted CIFAR10 Type 0 and Type 1 in Table 1. We again observe the superiority of our method regardless of the corruption types.

| Dataset | Ratio (%) | LfF [1] | Ours |
|---|---|---|---|
| Corrupted CIFAR-10 Type 0 | 0.5 | $33.95_{\pm 3.97}$ | $\mathbf{36.89}_{\pm 0.83}$ |
| | 1.0 | $41.54_{\pm 3.26}$ | $\mathbf{44.43}_{\pm 1.29}$ |
| | 2.0 | $50.45_{\pm 0.39}$ | $\mathbf{52.01}_{\pm 0.44}$ |
| | 5.0 | $58.99_{\pm 0.23}$ | $\mathbf{60.18}_{\pm 1.05}$ |
| Corrupted CIFAR-10 Type 1 | 0.5 | $35.07_{\pm 0.63}$ | $\mathbf{36.52}_{\pm 1.05}$ |
| | 1.0 | $42.32_{\pm 2.58}$ | $\mathbf{43.64}_{\pm 1.10}$ |
| | 2.0 | $49.05_{\pm 1.96}$ | $\mathbf{52.23}_{\pm 1.51}$ |
| | 5.0 | $58.77_{\pm 0.99}$ | $\mathbf{59.3}_{\pm 0.85}$ |

Table 1: Image classification accuracy evaluated on unbiased test sets of Corrupted CIFAR-10 Type 0 and Type 1 with varying ratio of bias-conflicting samples. Best performing results are marked in bold.

## B    Reconstruction of Disentangled Representation on BFFHQ

Fig. 1 supplements Fig. 4 of the main paper by showing the reconstructed images of disentangled latent vectors $z_i$ and $z_b$ on BFFHQ. Similar to Fig. 4 of the main paper, columns and rows correspond to those images where the bias attribute (*i.e.*, gender) and the intrinsic attribute (*i.e.*, age) are extracted, respectively. As mentioned in Section 5 of the main paper, we define 'age' as either 'young' or 'old' in our work. The latent vectors extracted from images of each column and row are concatenated to reconstruct their corresponding images, as shown in the middle. While we only utilize a decoder for the Colored MNIST trained with the reconstruction loss in Fig. 4 of the main paper, we also use a discriminator with an adversarial loss [3] to improve the quality of reconstructed images on BFFHQ.

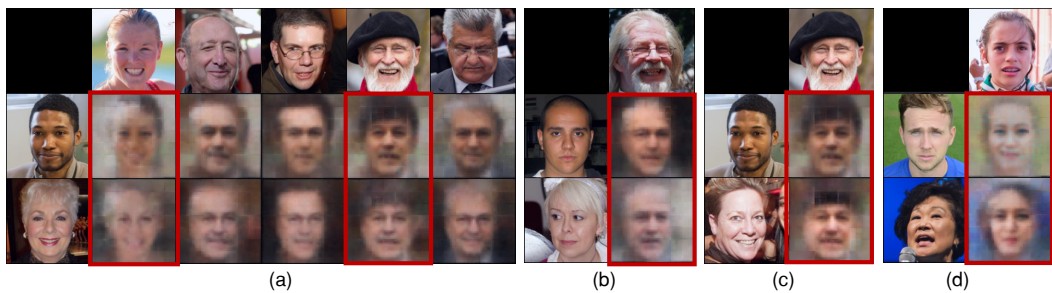

Figure 1: Reconstructed images from disentangled representations on BFFHQ. Columns and rows indicate those samples where the bias attribute (gender) and the intrinsic attribute (age) are extracted, respectively. By swapping the bias features with a given intrinsic feature, we observe that the gender changes while maintaining the age. In addition, by swapping the intrinsic features with a given bias feature, we change the ages while maintaining the gender. The reconstructed images in the highlighted red boxes indicate those samples with obvious age gaps, indicating that the latent vectors are properly disentangled.

The first row and the column of Fig. 1(a), (b), (c), and (d) indicate the images used to extract the latent vectors of the bias attribute (*i.e.*, gender) and the intrinsic attribute (*i.e.*, age), respectively. Genders of facial images on the leftmost column of Fig. 1(a), (b), (c), and (d) change according to the genders of faces on the top row. For example, in the first column of Fig. 1(a), the male in the second row changes to female in the second column while the female in the third row changes to male in the fifth column. In addition, we observe that the ages of reconstructed images change as the row changes. Note that the highlighted red boxes indicate the representative samples with clear age transitions as the row changes. In the first row of Fig. 1(a), the young female in the second column becomes old in the third row, while the old male in the fifth column becomes young in the second row. These examples verify that both $z_i$ and $z_b$ successfully contain the disentangled attributes for 'age' and 'gender' extracted from each image in the first columns and rows, respectively.

Similar to the decoder used in Colored MNIST, the decoder for BFFHQ is trained independently from our classification models. Due to this fact, the reconstructed samples may seem blurry or include images with less diversity. Table 3 illustrates the architecture we used to reconstruct the images of BFFHQ. We also provide training details of the decoder for BFFHQ in Section D.4.

## C   How disentanglement is guaranteed

Our proposed method includes three factors to guarantee the disentangled representations. First, ($E_b$ and $C_b$) are trained with the GCE loss to learn the 'easy-to-learn' attributes (i.e., bias attributes) from the images. In contrast, given the emphasized losses on the bias-conflicting samples by the GCE-based re-weighting method descibed in Eq. 2 of the main paper, ($E_i$ and $C_i$) learn the intrinsic attributes without being overfitted to the bias attributes. Second, the CE loss obtained from $C_i$ is not back-propagated into the $E_b$, and vice versa. This enables $E_b$ to not learn the intrinsic attributes, and vice versa. Third, for the representation $z_{\text{swap}} = [z_i; \tilde{z}_b]$, the classifier $C_i$ learns to predict the target label of the $z_i$ regardless of the $\tilde{z}_b$. On the other hand, the classifier $C_b$ is trained to predict the target label of the $z_{\text{swap}}$ regardless of the $z_i$. Again, this enforces $z_i$ and $z_b$ to be disentangled.

## D   Implementation Details

### D.1   Datasets

**Colored MNIST** This biased dataset consists of two highly correlated attributes, *color* and *digit*, following the existing literature [1, 4, 5, 6, 7]. We inject certain color into the foreground of each digit, following Nam *et al.* [1] and Darlow *et al.* [7]. We obtain the total images of bias-aligned samples and bias-conflicting samples for different ratios of bias-conflicting samples: (54,751, 249)-0.5%, (54,509, 491)-1%, (54,014, 986)-2%, and (52,551, 2,449)-5%.

**Corrupted CIFAR-10** We set the corruption types for Corrupted CIFAR-10 dataset in our paper as *Snow, Frost, Fog, Brightness, Contrast, Spatter, Elastic, JPEG, Pixelate,* and *Saturate,* among 15 different corruptions introduced in the original dataset [8]. These types of corruptions are highly correlated with the original classes of CIFAR-10 [9], which are *Plane, Car, Bird, Cat, Deer, Dog, Frog, Horse, Ship,* and *Truck.* Among five different severity of corruptions described in the original paper [8], we use the most severe level of corruptions for our dataset. Following are the total images of bias-aligned samples and bias-conflicting samples for each ratio of bias-conflicting samples: (44,832, 228)-0.5%, (44,527, 442)-1%, (44,145, 887)-2%, and (42,820, 2,242)-5%.

**BFFHQ** We compose the dataset by utilizing Flickr-Faces-HQ (FFHQ) Dataset [2] along with its various facial information, such as head pose and emotions. Among these features, we choose *age* and *gender* as two attributes with the strong correlation, as mentioned in Section 5 of the main paper. The dataset consists of 19,200 images for training (19,104 for bias-aligned and 96 for bias-conflicting), and 1,000 samples for test.

### D.2   Image Preprocessing

We train and evaluate our model with a fixed size of 28×28 and 32×32 images for Colored MNIST and Corrupted CIFAR-10, respectively, and 224×224 for BFFHQ.

| Part | Output shape | Layer Information |
|---|---|---|
| Input vector | $(B, 32)$ | – |
| Decoder | $(B, 512)$ | Linear$(32, 512)$, ReLU |
| | $(B, 1024)$ | Linear$(512, 1024)$, ReLU |
| | $(B, 3 * 28 * 28)$ | Linear$(1024, 3 * 28 * 28)$, ReLU |
| | $(B, 3 * 28 * 28)$ | Tanh |

Table 2: Decoder architecture used for reconstruction of images from disentangled latent vectors on Colored MNIST. These layers are composed in a reverse order of the MLP encoder used for Colored MNIST. B denotes the batch size.

| Part | Output shape | Layer Information |
|---|---|---|
| Input vector | $(B, 1024, 1, 1)$ | – |
| Decoder | $(B, 512, 7, 7)$ | ConvTrans$(1024, 512, K7, S2)$, ReLU |
| | $(B, 256, 14, 14)$ | ConvTrans$(512, 256, K3, S2, P1)$, ReLU |
| | $(B, 128, 28, 28)$ | ConvTrans$(256, 128, K3, S2, P1)$, ReLU |
| | $(B, 64, 56, 56)$ | ConvTrans$(128, 64, K3, S2, P1)$, ReLU |
| | $(B, 64, 112, 112)$ | Upsampling |
| | $(B, 3, 224, 224)$ | ConvTrans$(64, 3, K3, S2, P1)$ |

Table 3: Decoder architecture used for reconstruction of images from disentangled latent vectors on BFFHQ.

The images of Corrupted CIFAR-10 and BFFHQ are preprocessed with random crop and horizontal flip transformations, and also normalized along each channel $(3, H, W)$ with the mean of $(0.4914, 0.4822, 0.4465)$ and standard deviation of $(0.2023, 0.1994, 0.2010)$. For Colored MNIST, we do not use any augmentation techniques to preprocess the images.

### D.3 Training Details

For training, we utilize Adam [10] optimizer with default parameters (*i.e.*, betas $= (0.9, 0.999)$ and weight decay $= 0.0$) provided in PyTorch library. Learning rates of $0.01$ and $0.0001$ are used for training Colored MNIST and BFFHQ, respectively, and $0.0005$ for $0.5\%$ ratio of Corrupted CIFAR10 and $0.001$ for the remaining ratios of Corrupted CIFAR10. For each dataset, we use StepLR for learning rate scheduling. The decaying step is set to 10K for all datasets, and the decay ratio is set to $0.5$ for both Colored MNIST and Corrupted CIFAR10 and $0.1$ for BFFHQ. With the proposed scheduled feature augmentation, we start to schedule the learning rate after the feature augmentation was performed. For the proposed objective functions, we use a set of hyper-parameters $(\lambda_{\text{dis}}, \lambda_{\text{swap}_b}, \lambda_{\text{swap}})$ as $(10.0, 10.0, 1.0)$ for Colored MNIST and $(2.0, 2.0, 0.1)$ for BFFHQ, respectively. For Corrupted CIFAR10, we used $(5.0, 5.0, 1.0)$ for the ratio of $1\%$ and $2\%$, and $(1.0, 1.0, 1.0)$ for the ratio of $0.5\%$ and $5\%$. We conduct our experiments mainly using a single RTX 3090 gpu.

### D.4 Decoder for image reconstruction

This section provides a detailed explanation of the decoder used for reconstructing images from our disentangled latent vectors, described in Section 5.3 in the main paper and Section B. The architectures of the decoder for Colored MNIST and BFFHQ are shown in Tables 2 and 3, respectively. For Colored MNIST, we use the mean squared error for the reconstruction loss between the original and generated images, and Adam [10] optimizer with its learning rate of $0.001$. For BFFHQ, in addition to the reconstruction loss, we utilize adversarial loss, as proposed in LSGAN [11]. In this respect, we train a discriminator which has the same architecture as our encoder, *i.e.*, ResNet18 [12]. We use Adam optimizer with the learning rate of $0.001$ for training the decoder and the same optimizer with $0.0001$ for the discriminator, respectively. For every iteration, the decoder takes latent vectors extracted from our encoders as inputs and generates the reconstructed images as outputs. Since we utilize the decoder for the purpose of visualization, the losses of the decoder and the classification models are not backpropagated to each other.

# E  Broader Impacts and Limitations

As machine learning becomes a crucial part of our daily life in various forms of applications, it is crucial to validate the robustness and reliability of machine learning models. The dataset bias [13] causes the model to be susceptible to the peripheral features, rather than capturing the intrinsic features that humans usually rely on in image classification. This could raise a distrust issue of machine learning in various tasks sensitive to the safety concern [14, 15] or social equality [16, 17, 18]. Thus, as previous literature has pointed out [1, 4, 5, 6, 19, 20, 21, 22], it becomes important to build classification models that do not rely on bias attributes but rather learn the intrinsic attributes of a particular class.

Existing approaches address this issue by emphasizing bias-conflicting samples or suppressing the training of bias-aligned images, in order to avoid overfitting to the biased representations. However, as mentioned before, an extremely scarce number of bias-conflicting samples prevent the model from learning the generalizable intrinsic attributes of a certain class. In this paper, we propose an augmentation-based debiasing approach, fully utilizing a large proportion of bias-aligned features to diversify the visual features of bias-conflicting samples. Thus, we achieve the state-of-the-art debiasing performance on both synthetic and real-world datasets against existing baselines.

As a limitation, we acknowledge that learning *completely* disentangled representations by the proposed method remains challenging. The difficulty derives from the extreme scarcity of bias-conflicting samples and the highly correlated complex attributes in real-world images such as age and gender. Obtaining fully disentangled latent vectors for feature augmentations may further require hand-crafted modules for certain bias types, which is out of scope in this paper since we do not predefine a bias type in advance. Despite such a limitation, we believe that our approach provides a novel perspective of augmenting diversified bias-conflicting samples for learning debiased representations.