# OpenReview forum: "Learning Debiased Representation via Disentangled Feature Augmentation"
_NeurIPS.cc/2021/Conference — NeurIPS 2021 Oral_

### Official Review · Reviewer_3RPm · 2021-06-28

**Rating:** 7
**Confidence:** 5

**Summary:**

This paper proposed a feature-level augmentation method to disentangle the bias and intrinsic features for debiasing image classification. This paper pointed out that the diversity of augmented bias-conflicting (i.e., few-shot, tailed) data is crucial for debiasing. The proposed framework consists of two encoders that encode the image into disentangled intrinsic and bias features. (1) To achieve the disentanglement, the modules for intrinsic and bias features are trained using cross-entropy (CE) loss and generalized cross-entropy (GCE) loss, respectively. (2) To generate augmented data, the intrinsic and bias features are swapped in each mini-batch. Experiments conducted on two synthetic datasets Colored MNIST and Corrupted CIFAR-10) and two real-world datasets (BAR and Biased FFHQ) demonstrate that the proposed method achieves the best performance compared to those that do not require a bias type. The quantitative evaluation further illustrates the disentanglement.

**Limitations And Societal Impact:**

The authors have adequately addressed the limitations and potential negative societal impact of their work.

**Main Review:**

(+ Strengths, - Weaknesses, * Typos/Comments)

Overall, I like and value the research topic and motivation of this paper and lean positive. However, some details are not clear enough. I would update my rating depending on the authors' feedback. The details are as follows.

+ \+ Interesting and important research problem. This paper focuses on how to obtain disentangle representations for feature-level augmentation. This topic is interesting and important, and will attract many interests of the NeurIPS community.

+ \+ Good quality of writing and organization. Overall, the writing quality is good and the paper is well organized. It is comfortable to read this paper, although some details are not clear.

+ \+ Comprehensive experiments. Experiments are conducted on two synthetic datasets Colored MNIST and Corrupted CIFAR-10) and two real-world datasets (BAR and Biased FFHQ).

- \- Relative difficulty score and generalized cross-entropy (GCE) loss. It is not clear how the relative difficulty score $W(x)$ in Eq. (1) is used in the pipeline. W(x) is not mentioned again in both the overall objective functions Eq. (2) or Algorithm 1. Since readers may not be familiar with the generalized cross-entropy (GCE) loss, it is encouraged to briefly introduce the formulation and key points of the GCE loss to make this paper more self-contained.

- \- How bias-conflicting samples and bias-aligned samples are selected. This weakness follows the first one. It seems that the "bias-conflicting" is determined based on the relative difficulty score, but the details are missed. Also, the ablation study on how the "bias-conflicting" is determined, e.g., setting the threshold for the relative difficulty score, is encouraged to be considered and included.

- \- Disentanglement. It is not clear how disentanglement is guaranteed. Although "Broader Impacts and Limitations" stated that "Obtaining fully disentangled latent vectors ... a limitation", it is still important to highlight how the disentanglement is realized and guaranteed without certain bias types.

- \- Inference stage. It is not clear how the inference is conducted during testing. Which encoders/decoders are preserved during the test stage?

- \- Figure 1 is not clear. First, it seems that the two $y$ towards $L_{\text{CE}}$ are the outputs of $C_i$, but they are illustrated like labels rather than predictions. Second, the illustration of the re-weighting module is not clear. Does it represent Eq. (4)?

- \- Table 4 reported a much lower performance of "swapping" on BAR compared to the other three datasets. Is there any explanation for this, like the difference of datasets?

- \- Sensitivity to hyperparameters. The proposed framework consists of three important hyperparameters, $(\lambda_{\text{dis}},$ $\lambda_{swap_b},$ $\lambda_{\text{swap}})$. It is not clear whether the framework is sensitive to these hyperparameters and how these hyperparameters are determined.

* \* (Suggestion) Illustration of backpropagation. As introduced in  Line 167-168, the loss from $C_i$ is not backpropagated to $E_b$. It would be clearer if this can be added in Figure 1.

* \* Line 280. Is "the first row and column ... respectively" a typo? It is a little confusing for me to understand this.

* \* Typos in Algorithm 1. Are $\lambda_\text{dis}$ and $\lambda_{swap_b}$ missed in $L_{\text{dis}}$ and $L_{\text{swap}}$?

* \* Typo in Line 209. Corrputed -> Corrupted.





============================= After rebuttal ===================================

After reading the authors' response to my questions and concerns, I would like to vote for acceptance.

The major strengths of this paper are:

* The research problem, unbiased classification via learning debiased representation, is interesting and would attract the NeurIPS audience's attention.
* The proposed method is simple but effective. The method is built on top of LfF [12] and further considers (1) intrinsic and bias feature disentanglement and (2) data augmentation by swapping the bias features among training samples.
* The paper is clearly written and well organized.

These strengths and contributions are also pointed out by other colleague reviewers.

My main concerns were:

* Unclear technical details of the GCE loss and the relative difficulty score. This concern was also shared with Reviewer 8Ai1 and iKKw. The authors' response clearly introduced the details and addressed my concern well.

* Sensitivity to hyper-parameters. The authors' response provided adequate results to show the sensitivity to hyper-parameters.
Other details of implementation and analysis of experimental results. The authors' responses clearly answered my questions.

Considering both strengths and the weakness, I am happy to accept this paper.

**Time Spent Reviewing:**

4

---

> ### Author Response · Authors · 2021-08-10
> **Response to reviewer 3RPm (R4)**
>
> **GCE loss and relative difficulty score W**
>
> As the reviewer pointed out, the main paper mistakenly omits the multiplication of $W$ and the cross-entropy (CE) loss in Eqs. (2) and (3).
>
> The revised Eqs. (2) and (3) are as follows:
>
> $L_\text{dis} = W(x)\cdot CE(C_i(z), y) + \lambda_\text{dis} GCE(C_b(z), y)$,
>
> $L_\text{swap} = W(x)\cdot CE(C_i(z_\text{swap}), y) + \lambda_{\text{swap}_b} GCE(C_b(z_\text{swap}), \tilde{y})$.
>
> The Generalized Cross Entropy (GCE) loss [1] is described as
>
> $GCE(p(x), y) = \frac{1-p(x)^{q}}{q}$,
>
> where $y$ refers to the ground truth label, $p(x)$ indicates the softmax output of the neural network for an image $x$, and $q\in(0,1)$ is a hyper-parameter.
> Compared to the CE loss, the GCE loss imposes high weights on the gradients for the samples which have high probability of the target class $y$.
>
> Therefore, when training models with the GCE loss in a biased dataset, it emphasizes the training on the easy samples (i.e., bias-aligned samples) with the high values of probability for the target label, leading to the network fully biased.
> In this respect, we train the ($E_{b}$ and $C_{b}$) with the GCE loss, and ($E_{i}$ and $C_{i}$) with the CE loss.
> As ($E_{b}$ and $C_{b}$) are trained to be fully biased, we can obtain the high CE loss when the bias-conflicting samples are given to the ($E_{b}$ and $C_{b}$), while having the relatively small CE loss with the ($E_{i}$ and $C_{i}$).
> Utilizing the Eq. (1) in the main paper, we can obtain the high relative difficulty score $W$ for the bias-conflicting samples.
> Therefore, according to the revised Eqs. (2) and (3), the CE loss re-weighted with the $W$ can emphasize the training on the bias-conflicting samples and thus enables the learning of intrinsic attributes.
>
> **Selecting bias-aligned/conflicting samples**
>
> Through the revised Eqs. (2) and (3) on $W$ and the GCE loss, samples with relatively large $W$ can be regarded as more bias-conflicting compared to the ones with small $W$. Therefore, the bias-conflicting samples are defined "relatively" by $W$, instead of the explicit threshold, in our method.
> We will include such additional explanation on how the bias-conflicting samples are determined via $W$ in the camera ready.
>
> **Guarantee of disentanglement**
>
> Our proposed method includes three factors to achieve the disentangled representations.
> First, ($E_b$ and $C_b$) are trained with the GCE loss to learn the ‘easy-to-learn’ attributes (i.e., bias attributes) from the images. In contrast, given the emphasized losses on the bias-conflicting samples by the GCE-based re-weighting method, ($E_i$ and $C_i$) learn the intrinsic attributes without being overfitted to the bias attributes.
> Second, as mentioned in line 167-168, the CE loss obtained from $C_i$ is not back-propagated into the $E_b$, and vice versa. This enables $E_b$ to **not** learn the intrinsic attributes, and vice versa.
> Third, for the representation $z_{swap}=[z_i; \tilde{z_b}]$, the classifier $C_i$ learns to predict the target label of the $z_i$ regardless of the $\tilde{z_b}$. On the other hand, the classifier $C_b$ is trained to predict the target label of the $\tilde{z_b}$ regardless of the $z_i$. Again, this enforces $z_i$ and $z_b$ to be disentangled.
>
> **Clarification on inference stage**
>
> We use encoders $E_i$, $E_b$ and linear classifier $C_i$, and do not utilize $C_b$ during the test phase. To be more specific, we extract $z_i$ and $z_b$ by inputting an image $x$ into $E_i$ and $E_b$, respectively. Then, we concatenate $z=[z_i; z_b]$ and forward to $C_i$ for predicting the target labels. We thank R4 for pointing out such an important detail for the reproducibility of our work and we will include it in our camera ready.
>
> **Clarification on Fig. 1**
>
> As stated in the manuscript (line 164) of the main paper, we utilize the notation $y$ as the ground-truth target label in Fig. (1). The notation of the prediction is implicitly represented with the arrow coming from the linear classifier $C_i$. The prediction and the ground-truth label $y$ are used to calculate the CE loss and the GCE loss, denoted as $L_{CE}$ and $L_{GCE}$, respectively.
>
> We clarify that the illustration of the re-weighting module does not represent Eq. (4).
> Instead, it indicates the operation in Eqs. (2) and (3).
> REB-FIG6 shows the revised illustration of the re-weighting module in the Fig. (1) of the main paper.
> Our re-weighting module $R$ in REB-FIG6 includes the two-step operations: 1) As described in Eq. (1), it calculates the relative difficulty score $W$ given the CE losses from the two different pairs of an encoder and a linear classifier, i.e., ($E_i$, $C_i$) and ($E_b$, $E_b$), given an input image $x$. 2) Using the revised version of Eqs. (2) and (3), given the score, the re-weighting module calculates the $L_{dis}$ when given the feature $z=[z_i;z_b]$ (light-orange background in Fig. (1)), or $L_{dis}$ given the feature $z_{swap}=[z_i; \tilde{z_b}]$ (dark-orange background in Fig. (1)).
> We highlighted the revised parts by a red dotted box.
>
> REB-FIG6 (Revised Fig. 1):
> https://tinyurl.com/4037-REBFIG6
>
> **Low performance of swapping in Table 4 on BAR**
>
> The low accuracy of "swapping" (evaluated with $z_{swap}=[z_i; \tilde{z_b}]$) compared to "original" (evaluated with $z=[z_i;z_b]$) of the BAR dataset in Table (4) may indicate that there may exist noisy samples in the augmented $z_{swap}$ in BAR dataset.
> In other words, while $z$ uses the intrinsic and bias attributes extracted from the same image, $z_{swap}$ is augmented by the attribute pairs from two different images.
> Such noisy augmentation may be due to the imperfect disentangled representations when our model is trained on the BAR dataset.
> We conjecture that one of the reasons for such imperfect disentanglement is the limited number of training images in the BAR dataset.
> More specifically, compared to Colored MNIST, Corrupted CIFAR10, and BFFHQ which include approximate 50K, 44K, and 19K images for the training set, respectively, the BAR dataset contains less than 2K images. This makes it more challenging for the model to learn fully disentangled representation compared to other datasets in a limited amount of training images.
>
> As mentioned in Section E in the supplementary, we acknowledge that such factor can limit the learning of completely disentangled representation.
> Still, our method maintains the classification ability on the original features obtained from the unbiased test set.
> We will add such a detailed explanation on the performance gap between original and swapping features in Table (4).
>
> **Sensitivity to hyper-parameters**
>
> We demonstrate additional results of networks trained with various values of the three hyper-parameters: $\lambda_{dis}$, $\lambda_{swap_{b}}$ and $\lambda_{swap}$.
> We tuned the values of hyper-parameters we utilized in the paper (supplementary Section D.3).
> Since there are three hyper-parameters, we changed one hyper-parameter by multiplying it with $0.25$, $0.5$, $2$, and $4$ at a time while preserving the other two unchanged.
> Then, we iterate the same procedure for the other hyper-parameters in turn. For every dataset except BFFHQ, we used the ratio of bias-conflicting samples of 1%. For BFFHQ, we used the ratio of 0.5%. The first column and row indicate the hyper-parameter which was changed and the multiplied value (i.e., scale), respectively. For each row, the last column presents the standard deviation of the accuracies shown in the second to the sixth column.
> Table (1) shows that our approach is fairly robust against the varied hyper-parameters, considering the reasonably low values of standard deviation, denoted as stdv.
>
> #### **Table 1**
>
> | Dataset\scale |  0.25 |  0.5 |  1 |  2 |  4 | stdv |
> | :---: | :---: |:---: |:---: | :---: |:---: |:---: |
> | Colored MNIST | | | | | | |
> | $\lambda_{dis}$| 80.07±2.02 | 79.85±0.64 | 81.73±2.34 | 79.58±2.06 | 80.40±0.77 | 0.84 | |
> | $\lambda_{swap_{b}}$ | 79.35±0.83 | 78.45±1.26 | 81.73±2.34 | 78.32±0.77 | 79.64±1.40 | 1.37 |
> | $\lambda_{swap}$ | 80.70±1.43 | 77.31±2.09 | 81.73±2.34 | 77.21±3.57 | 80.08±2.32 | 2.04 |
> | Corrupted CIFAR10| | | | | | |
> | $\lambda_{dis}$| 35.43±1.16 | 31.81±4.65 | 36.49±1.79 | 34.07±1.69 | 34.56±2.36 |  1.74 |
> | $\lambda_{swap_{b}}$ | 32.27±4.57 | 34.12±3.16 | 36.49±1.79 | 34.96±2.15 | 33.88±3.36 | 1.54 |
> | $\lambda_{swap}$ | 34.78±1.81 | 35.75±1.98 | 36.49±1.79 | 35.33±0.87 | 35.54±4.32 | 0.62 |
> |BAR| | | | | | | |
> | $\lambda_{dis}$| 52.14±3.99 | 51.20±0.62 | 52.31±1.00 | 53.90±0.44 | 53.72±1.36 |  1.13 |
> | $\lambda_{swap_{b}}$ | 54.6±2.57 | 51.38±1.07 | 52.31±1.00 | 52.38±1.53 | 53.49±0.62 | 1.23 |
> | $\lambda_{swap}$ | 54.85±2.17 | 51.43±3.52 | 52.31±1.00 | 54.43±2.48 | 53.66±2.10 | 1.43 |
> |BFFHQ| | | | | | |
> | $\lambda_{dis}$| 62.53±2.02 | 63.47±2.40 | 63.87±0.31 | 61.73±1.33 | 61.2±2.03 | 1.12 |
> | $\lambda_{swap_{b}}$ | 60.8±2.43 | 62.27±2.97 | 63.87±0.31 | 62.8±1.25 | 60.67±1.27 | 1.35 |
> | $\lambda_{swap}$ | 59.87±1.51 | 60.00±0.2 | 63.87±0.31 | 64.4±4.59 | 60.87±2.00 | 2.17 |
>
> **Illustration of back-propagation in Fig. (1)**
>
> We deeply appreciate the suggestion regarding the Fig. (1). We will consider updating the Fig. (1) including the illustration of the back-propagation.
>
> **Line 280. Is "the first row and column ... respectively" a typo?**
>
> The images in the first row indicate the images used for extracting the bias attributes (i.e., color), while the ones in the first column are the images for extracting the intrinsic attributes (i.e. digit).
>
> **Typos in Algorithm 1 and Line 209.**
>
> We will include the missing lambda values in Algorithm 1 and fix the typo in Line 209, thanks to R4.
>
> [1] [Z. Zhang and M. Sabuncu. Generalized cross entropy loss for training deep neural networks with noisy labels. In Advances in Neural Information Processing Systems, 2018.](https://papers.nips.cc/paper/2018/file/f2925f97bc13ad2852a7a551802feea0-Paper.pdf)

---

> > ### Comment · Reviewer_3RPm · 2021-08-28
> > **Happy to increase my score.**
> >
> > Thank the authors for their effort and time during the rebuttal. After reading the authors' response to my questions and concerns, I would like to vote for acceptance.
> >
> > The major strengths of this paper are:
> >
> > 1. The research problem, unbiased classification via learning debiased representation, is interesting and would attract the NeurIPS audience's attention.
> >
> > 2. The proposed method is simple but effective. The method is built on top of LfF [12] and further considers (1) intrinsic and bias feature disentanglement and (2) data augmentation by swapping the bias features among training samples.
> >
> > 3. The paper is clearly written and well organized.
> >
> > These strengths and contributions are also pointed out by other colleague reviewers.
> >
> > My main concerns were:
> >
> > 1. Unclear technical details of the GCE loss and the relative difficulty score. This concern was also shared with Reviewer 8Ai1 and iKKw. The authors' response clearly introduced the details and addressed my concern well.
> >
> > 2. Sensitivity to hyper-parameters. The authors' response provided adequate results to show the sensitivity to hyper-parameters.
> >
> > 3. Other details of implementation and analysis of experimental results. The authors' responses clearly answered my questions.
> >
> > Considering both strengths and the weakness, I am happy to accept this paper.

---

### Official Review · Reviewer_ub8A · 2021-07-10

**Rating:** 8
**Confidence:** 4

**Summary:**

The paper presents a way to generate bias-conflicting samples at feature level, which help de-bias the model. For this, the authors first propose learning two separate (disentangled) representations: a) representation of implicit attributes (i.e., necessary to predict the target) learned through the normal cross-entropy loss and b) representation of biased attributes learned through generalized cross entropy loss (also used by LfF for bias amplification). The empirical results demonstrate that generating such samples outperforms oversampling/reweighting.

**Limitations And Societal Impact:**

The paper acknowledges the imperfect disentanglement, but could also mention that the work does not study more realistic cases where biases can stem from different sources and the correlations between bias attributes and target labels may vary, which may pose greater difficulty for disentanglement.

**Main Review:**

The proposed idea is a very neat adaptation of the code swapping technique [1] applied for bias mitigation. This is a good contribution to the field, as it provides a simple way to generate bias-conflicting samples at the representation level without relying on generative models or annotations for the bias attributes/labels.

Overall the paper is very well written and properly structured. I like the upfront analysis on ColoredMNIST and CIFAR10, motivating the need to for diversity. Ablations and comparisons against other methods e.g., LfF clearly demonstrate that training with the bias-conflicting samples generated in the proposed way improves over simply up-weighting the bias-conflicting/down-weighting the bias-aligned samples.

The 2D visualizations and reconstructed samples do a good job of showcasing the disentanglement (though not always perfect) between the intrinsic/bias attributes.

Next, I have a few comments/suggestions:

[C1] The main weakness is that the benchmarks used in the work seem to contain only one type of spurious correlation. I wonder what happens if there are multiple types of biases. Would the same GCE-based biased model capture all of those biases? Would one need to create multiple bias models and tune the GCE loss individually to capture those biases?

[C2] Following up on [C1], it would have been a huge plus to have stronger benchmarks with different kinds/levels of biases e.g., GQA-OOD [2].

[C3] Fig. 3 was a bit confusing and could be labeled/captioned better. I think both rows (target type) and columns (representation type) can be labeled to make things clearer. The BAR plot was a bit confusing -- I suppose the bottom right for column ‘c’ is all green due to the unavailability of bias labels?

Other Minor comments:

[MC1] I would prefer the phrase: ‘intrinsic attributes’ to be replaced by something more obvious e.g., ‘signal’ or ‘task-relevant attributes’ or perhaps ‘non-spurious attributes’. However, this is not a major concern, since it is well defined in the abstract/introduction.

[MC2] Line 175/Algorithm 1: The phrase ‘Randomly shuffled bias attributes’ was slightly confusing, as it could also mean shuffling the dimensions within the representation itself as opposed to swapping the entire ‘code’ between samples. Perhaps the text could say something like: "replace $z_b$ in $z$ with  $\tilde{z_b}$ from a randomly chosen sample in the batch to create the swapped representation: $z_{swap}$ ..."


Having said this, all in all, this is a good paper, with potential for high practical impact.

[1] Park, Taesung, et al. "Swapping autoencoder for deep image manipulation." arXiv preprint arXiv:2007.00653 (2020).

[2] Kervadec, Corentin, et al. "Roses Are Red, Violets Are Blue... but Should Vqa Expect Them To?." Proceedings of the IEEE/CVF Conference on Computer Vision and Pattern Recognition. 2021.

**Time Spent Reviewing:**

4

---

> ### Author Response · Authors · 2021-08-10
> **Response to reviewer ub8A (R3)**
>
> **Resolving multiple types of bias**
>
> To verify the debiasing capability in a dataset with multiple types of bias, we constructed a toy-set dataset which consists of MNIST images correlated with two different bias types; color and corruption.
> To be specific, we applied a certain corruption type introduced in Hendrycks and Dietterich [1] to each class of the Colored MNIST dataset, making the strong correlation between digits and 1) colors and 2) corruption types.
> We utilized the corruption types which were used in 'Corrupted CIFAR10 type 0' in LfF [2].
> REB-FIG5 shows the examples of these data samples termed as 'Corrupted & Colored MNIST'.
> Afterwards, we trained both our proposed method and vanilla network on such dataset and evaluated them on the unbiased test dataset.
> Table (1) demonstrates that our method is still superior to the vanilla network when evaluated with Corrupted & Colored MNIST.
> As the GCE loss encourages the model to overly rely on the shortcut features, it can capture the biases as long as they are 'easy-to-learn'.
> Therefore, for the Corrupted & Colored MNIST, we believe that only a single GCE-based biased encoder in our method can even capture multiple biases, achieving the reasonable debiasing capability.
> The suggestion of tuning the multiple bias models for each bias type would be a promising solution, and we will further discuss in the camera ready.
>
> #### **Table 1**
> #### **[Corrupted & Colored MNIST]**
>
> | Diversity ratio \ Model | Vanilla | Ours |
> | :---: | :---: |:---: |
> | 0.5% | 34.74±3.05 | **52.65**±2.33 |
> | 1% | 44.07±0.62 | **61.14**±1.67 |
> | 2% | 62.24±1.73 | **70.20**±3.68 |
> | 5% | 74.97±0.27 | **76.64**±0.83 |
>
> REB-FIG5 (Example images of Corrupted & Colored MNIST):
> https://tinyurl.com/4037-REBFIG5
>
> **Evaluation on stronger benchmarks**
>
> We agree that learning the debiased representation on the dataset with diverse kinds of biases has significant importance in this field.
> We tried to adopt our method on the suggested dataset (e.g., GQA-OOD) but the one-week long rebuttal period was too short for us to conduct experiments on the novel domain of visual-question answering (VQA).
> Therefore, we will add discussion on such issue in the camera ready.
>
>
> **Clarification on Fig. (3)**
>
> The column (a) and (b) demonstrate the 2D projection of ‘intrinsic attributes ($z_i$)’ and that of ‘bias attributes ($z_b$)’ of Colored MNIST, respectively. The column (c) indicates the 2D projection of ‘intrinsic attributes ($z_i$)’ of BAR. The row (i) and (ii) include the projected latent vectors colorized with the ‘target labels’ and ‘bias labels’, respectively. Thanks to R3, we will clarify the notations and captions of Fig. (3) used in the current version. We will replace unclear notations (e.g., (a) and (i)) with straightforward notations (e.g., target attributes and target labels).
>
> As R3 mentioned, the bottom right for column (c) is all green due to the unavailability of bias labels.
> Therefore, we first utilized the $K$-means clustering on the $z_b$ vectors and obtain their cluster assignments. Then, we colorize them by utilizing such cluster assignments as the bias labels, as shown in REB-FIG1.
> Afterwards, to verify that the data samples are grouped according to the bias factor, we randomly sampled the images from each cluster and provide them in REB-FIG2.
> We found that the images from each cluster share the similar backgrounds/colors, even with different target labels (i.e., action).
> For example, the six images in the upper left corner of REB-FIG2 have different target labels but they all contain a grass or green background. Similarly, other sets of images are also clustered according to the backgrounds (e.g., ice/white, dirt/brown, and indoor/dark). Through the additional qualitative analysis, we observe that $z_b$ mainly contains the bias attributes, demonstrating that our method disentangles the intrinsic and bias attributes.
>
> REB-FIG1 (2D projection of $z_b$ for BAR): https://tinyurl.com/4037-REBFIG1
>
> REB-FIG2 (Representative images of each cluster): https://tinyurl.com/4037-REBFIG2
>
> **Suggestions on terminologies and notations**
>
> We appreciate the suggestion and will definitely change the notation to be more obvious in our camera ready.
>
> [1] [Hendrycks and Dietterich. Benchmarking Neural Network Robustness to Common Corruptions and Perturbations. In International Conference on Learning Representations, 2019](https://openreview.net/forum?id=HJz6tiCqYm)
>
> [2] [Junhyun Nam, Hyuntak Cha, Sungsoo Ahn, Jaeho Lee, and Jinwoo Shin. Learning from failure: Training debiased classifier from biased classifier. In Advances in Neural Information Processing Systems, 2020.](https://proceedings.neurips.cc/paper/2020/file/eddc3427c5d77843c2253f1e799fe933-Paper.pdf)

---

### Official Review · Reviewer_iKKw · 2021-07-13

**Rating:** 7
**Confidence:** 5

**Summary:**

Dataset bias significantly degrades the generalization performance of deep neural networks. To alleviate this problem, the authors propose a novel approach for developing the debiased representation, where the model learns the disentangled representation and synthesizes diverse bias-conflicting samples via feature-level data augmentation. They demonstrate its effectiveness through extensive experiments on the image classification task.


**Limitations And Societal Impact:**

yes.

**Main Review:**

[Strengths]

The idea is simple and interesting. They propose a straightforward and effective debiasing approach via disentangled feature augmentation. Also, the proposed method handles debiasing without presuming a certain bias type, which is advantageous for a practical setting.
They synthesize diverse bias-conflicting samples with feature-level data augmentation. And via a toy-set experiment, they show that the diversity of bias-conflicting samples is an important factor in debiasing.
They demonstrate the effectiveness of the proposed method through extensive experiments on the image classification task with synthetic and real-world datasets. (i.e., Colored MNIST, Corrupted CIFAR-10, BAR and bFFH)


[Weakness]

It is questionable whether the experiment on LfF [1] was performed properly. There is a significant gap between the reproduced performance of LfF in Table 2 and the reported performance in LfF. On corrupted CIFAR-10, the reported performance in LfF outperforms the proposed method. For validating a fair comparison, the authors should clarify this.

EnD [2] also handles debiasing based on the disentangled representation. Thus, at least the discussion with EnD is necessary; (conceptually and if possible empirically) comparing  the proposed method with it.

To show the effectiveness of debiasing, it should evaluate the performance on bias-conflicting samples as LfF [1] and EnD [2]. However, evaluations and comparisons are conducted only on unbiased samples in experimental results. By observing and evaluating on bias-conflicting samples, it would be clear where the effectiveness of the proposed method comes from.

Unlike LfF [1], the objective function of Equation (2) does not include the relative difficulty score W. Hence, it is not clear how E_i and C_i can learn intrinsic attributes. It should be clearly explained whether missing W in Eq (2) has any meaning (or reason)-- if so, how E_i and C_i can learn intrinsic attributes.

[Minor]

For readability, it would be better to add explanations for (i) and (ii) in Figure 3.
It would be great to show the effect of the proposed method by visualizing before and after applying the proposed method using CAM (or Grad-CAM) in the real-world dataset, BAR.
For example, in a bias-aligned sample such as ‘birds on the sky’, the proposed method may lead the model focus on the birds, not the sky.

[1] Junhyun Nam, Hyuntak Cha, Sungsoo Ahn, Jaeho Lee, and Jinwoo Shin. “Learning from failure: Training debiased classifier from biased classifier.” In Advances in Neural Information Processing Systems, 2020.

[2] Tartaglione, Enzo, Carlo Alberto Barbano, and Marco Grangetto. "EnD: Entangling and Disentangling deep representations for bias correction." Proceedings of the IEEE/CVF Conference on Computer Vision and Pattern Recognition. 2021.


**Time Spent Reviewing:**

5

---

> ### Author Response · Authors · 2021-08-10
> **Response to reviewer iKKw (R2)**
>
> **Reproducibility of LfF [1]**
>
> The main reason for the gap between the performance of LfF in our paper and the original paper [1] lies in the different experimental setup.
> First, for the Corrupted CIFAR10, we utilized the ResNet-18 architecture while LfF used the ResNet-20.
> In our paper, the main reason for choosing the ResNet-18 for the Corrupted CIFAR10 was to maintain the consistency in network architectures across the other datasets including BAR and BFFHQ. Another reason for the performance difference is that the Corrupted CIFAR10 dataset in our paper is constructed with the different types of corruptions compared to the ones in the original LfF paper.
> While we randomly sampled 10 corruptions among 20 types, LfF constructed two sets of Corrupted CIFAR10 with different corruption types, which were termed as 'type 0' and 'type 1'.
>
> To demonstrate that our implementation of LfF was correct, we newly constructed the same sets of Corrupted CIFAR10 datasets used in LfF. Afterwards, we trained the ResNet-20 network with these datasets and report the classification accuracies in Table (1). It is observed that the test accuracies on the unbiased test sets of our implementation are similar to the ones in LfF paper.
>
> #### **Table 1**
>
> #### **[Type0]**
> | Diversity ratio | Original Paper | Reproduced  |
> | :---: | :---: |:---: |
> | 0.5% | 31.66±1.18 | 33.95±3.97 |
> | 1% | 41.37±2.34 | 41.54±3.26 |
> | 2% | 49.43±0.78 | 50.45±0.39 |
> | 5% | 59.95±0.16 | 58.99±0.23 |
>
> #### **[Type1]**
> | Diversity ratio | Original Paper | Reproduced  |
> | :---: | :---: |:---: |
> | 0.5% | 34.11±2.39 | 35.07±0.63 |
> | 1% | 41.29±2.08 | 42.32±2.58 |
> | 2% | 48.75±1.68 | 49.05±1.96 |
> | 5% | 58.57±1.18 | 58.77±0.99 |
>
> Also, Table (2) compares our approach with LfF using the experimental settings reported in the original paper of LfF, i.e., ResNet-20 and Corrupted CIFAR10 of types 0 and 1.
>
> We again observe that our method robustly outperforms it in every dataset with various ratios of bias-conflicting samples for both types of corruptions.
>
> #### **Table 2**
>
> #### **[Type0]**
>
> | Diversity ratio | LfF | Ours |
> | :---: | :---: |:---: |
> | 0.5% | 33.95±3.97 | **36.89**±0.83 |
> | 1% | 41.54±3.26 | **44.43**±1.29 |
> | 2% | 50.45±0.39 | **52.01**±0.44 |
> | 5% | 58.99±0.23 | **60.18**±1.05 |
>
> #### **[Type1]**
>
> | Diversity ratio | LfF | Ours |
> | :---: | :---: |:---: |
> | 0.5% | 35.07±0.63 | **36.52**±1.05 |
> | 1% | 42.32±2.58 | **43.64**±1.10 |
> | 2% | 49.05±1.96 | **52.23**±1.51 |
> | 5% | 58.77±0.99 |**59.3**±0.85 |
>
>
> **Comparison with EnD [2]**
>
> EnD [2] proposes a simple but effective method which disentangles the representations belonging to the same bias class while entangling the ones belonging to the same target class.
> We deeply appreciate for suggesting this work and will add and cite this paper as our baseline in the camera ready.
>
> The main contribution of our method is that we do not require any prior knowledge on the bias type as well as the explicit supervision on the bias. Thus, this improves the generalization capability of our debiasing approach in the circumstances where acquiring such knowledge or the labels are hardly accessible. In contrast, to the best of our knowledge, EnD requires manually annotated bias labels beforehand which is labor-intensive and may even lack applicability on the real-world datasets where obtaining the bias labels is demanding in nature (e.g., BAR does not have bias labels).
> Therefore, we believe that our approach has wide applicability in addressing the debiasing task compared to EnD.
>
> We also compare the unbiased test accuracies of the two methods using the Colored MNIST, Corrupted CIFAR10, and BFFHQ datasets, where the bias labels are available. We could not evaluate EnD on the BAR dataset since BAR does not have bias labels. We follow the same implementation details illustrated in our main paper (Section 5.1), such as the network architecture and the dataset setup, when training the EnD for a fair comparison. Table (3) demonstrates that our method achieves the superior debiasing capability against the EnD on every dataset with various diversity ratios. One possible reason for the result is that EnD still relies on the limited number of bias-conflicting samples, while we diversify them via disentangled feature-level augmentation.
>
> #### **Table 3**
>
> #### **[Colored MNIST]**
>
> | Diversity ratio | EnD | Ours |
> | :---: | :---: |:---: |
> | 0.5% | 34.28±1.20 | **65.22**±4.41 |
> | 1% | 49.50±2.51 | **81.73**±2.34 |
> | 2% | 68.45±2.16 | **84.79**±0.95 |
> | 5% | 81.15±1.43 |**89.66**±1.09 |
>
> #### **[Corrupted CIFAR10]**
>
> | Diversity ratio | EnD | Ours |
> | :---: | :---: |:---: |
> | 0.5% | 22.89±0.27 | **29.95**±0.71 |
> | 1% | 25.46±0.41 | **36.49**±1.79 |
> | 2% | 31.31±0.35 | **41.78**±2.29 |
> | 5% | 40.26±0.85 |**51.13**±1.28 |
>
> #### **[BFFHQ]**
>
> | Diversity ratio | EnD | Ours |
> | :---: | :---: |:---: |
> | 0.5% | 56.87±1.42 | **63.87**±0.31 |
>
> **Evaluation on bias-conflicting samples**
>
> We present the classification accuracies evaluated on the bias-conflicting test set for our method and other existing baselines in Table (4).
> Note that for BAR and BFFHQ, the results reported in the main paper are evaluated on the bias-conflicting test set, following the same evaluation protocol in LfF.
> Therefore, we newly constructed the bias-conflicting test set for Colored MNIST and Corrupted CIFAR10 by excluding the bias-aligned test samples from the unbiased test set we utilized in the main paper. We observe that our method still outperforms existing baselines when evaluated on the bias-conflicting samples only.
> This is another important evaluation result that we will add in the camera ready.
> Bold and italic numbers indicate the best and second-best accuracy, respectively.
>
> #### **Table 4**
>
> #### **[Colored MNIST]**
>
> | Diversity ratio | Vanilla | HEX | ReBias | LfF | Ours |
> | :---: | :---: | :---: | :---: | :---: | :---: |
> | 0.5% | 28.26±0.34 | 21.72±2.14 | **69.12**±1.14 | 52.64±9.22 | *62.53*±3.87 |
> | 1% | 46.04±2.45 | 34.18±2.78 | **85.04**±0.55 | 65.70±5.93 | *81.26*±0.10 |
> | 2% | 62.97±3.52 | 53.58±3.40 | **91.83**±0.68 | 74.19±4.20 | *82.65*±1.13 |
> | 5% | 81.49±2.46 | 69.19±3.74 | **96.68**±0.12 | 81.70±3.91 | *88.27*±1.89 |
>
> #### **[Corrupted CIFAR10]**
>
> | Diversity ratio | Vanilla | HEX | ReBias | LfF | Ours |
> | :---: | :---: | :---: | :---: | :---: | :---: |
> | 0.5% | 16.06±0.66 | 11.17±0.05 | 15.22±0.64 | *25.01*±0.72 | **26.42**±1.65 |
> | 1% | 16.36±0.38 | 11.51±0.31 | 17.46±0.29 | *27.93*±1.14 | **29.76**±1.32 |
> | 2% | 20.22±0.27 | 12.35±0.39 | 25.06±0.94 | *35.92*±0.54 | **36.87**±1.10 |
> | 5% | 29.09±0.25 | 13.19±0.21 | 37.35±0.60 | *49.33*±1.17 | **49.62**±0.51 |
>
> **GCE loss and relative difficulty score W**
>
> As the reviewer pointed out, the main paper mistakenly omits the multiplication of $W$ and the cross-entropy (CE) loss in Eqs. (2) and (3).
>
> The revised Eqs. (2) and (3) are as follows:
>
> $L_\text{dis} = W(x)\cdot CE(C_i(z), y) + \lambda_\text{dis} GCE(C_b(z), y)$,
>
> $L_\text{swap} = W(x)\cdot CE(C_i(z_\text{swap}), y) + \lambda_{\text{swap}_b} GCE(C_b(z_\text{swap}), \tilde{y})$.
>
> We will revise the equations in our camera ready.
>
> **Explanations in Fig. (3) and additional qualitative analysis**
>
> We will replace the unclear notations, such as (a) and (i), with more straightforward labels, such as target attributes and target labels, as suggested by R2. In compliance with the grateful suggestion, REB-FIG4 compares the visualization of our approach and the vanilla network using the Grad-CAM [3] on the BAR dataset.
> To this end, we utilized the pre-trained models of our method and vanilla which were trained at the time of submission of our main paper.
> By following the guideline of a widely used Github repository [4], we chose the last layer of the fourth convolutional block in ResNet-18 as the target layer to compute the Grad-CAM. REB-FIG4 shows 1) the original image, 2) Grad-CAM of ours, and 3) that of the vanilla network in each column. We observe that the regions which are highly activated on the images correspond to the semantically meaningful objects in our result, while the vanilla network mainly attends the background regions which may mainly work as the bias attributes. For example, for the “Throwing” category, the highlighted regions of the Grad-CAMs of our method generally indicate the ballplayer, while those of the vanilla model often are the sports fields. Similar results are also observed in other classes. We will add the results of Grad-CAM in the camera ready.
>
> REB-FIG4 (Grad-CAM for BAR): https://tinyurl.com/4037-REBFIG4
>
> [1] [Junhyun Nam, Hyuntak Cha, Sungsoo Ahn, Jaeho Lee, and Jinwoo Shin. Learning from failure: Training debiased classifier from biased classifier. In Advances in Neural Information Processing Systems, 2020.](https://proceedings.neurips.cc/paper/2020/file/eddc3427c5d77843c2253f1e799fe933-Paper.pdf)
>
> [2] [Enzo Tartaglione, Carlo Alberto Barbano, and Marco Grangetto. EnD: Entangling and Disentangling deep representations for bias correction. In Proceedings of the IEEE/CVF Conference on Computer Vision and Pattern Recognition (CVPR), 2021.](https://openaccess.thecvf.com/content/CVPR2021/papers/Tartaglione_EnD_Entangling_and_Disentangling_Deep_Representations_for_Bias_Correction_CVPR_2021_paper.pdf)
>
> [3] [Ramprasaath R Selvaraju, Michael Cogswell, Abhishek Das, Ramakrishna Vedantam, Devi Parikh, and Dhruv Batra. Grad-cam: Visual explanations from deep networks via gradient-based localization. In Proceedings of the IEEE international conference on computer vision, pages 618–626, 2017.](https://openaccess.thecvf.com/content_ICCV_2017/papers/Selvaraju_Grad-CAM_Visual_Explanations_ICCV_2017_paper.pdf)
>
> [4] https://github.com/jacobgil/pytorch-grad-cam

---

> > ### Comment · Reviewer_iKKw · 2021-08-14
> > **Response to authors' rebuttal**
> >
> > 1) Reproducibility  of LfF[1]
> > - The authors explained the discrepancy in reproduced results as the architecture & different type of corruption. Table 1 and Table 2 also show that the reproduced results using the same experimental setting as LfF[1], conducting the fair comparison. Therefore, my main concerns on experimental validation and fair comparison issue have been resolved after the authors' rebuttal.
> >
> > 2) Comparison with EnD: The authors explained the difference between EnD and the proposed method (requiring prior knowledge). In addition, Table 3 also demonstrates that the proposed method outperforms EnD[2].
> >
> > 3) Absence of evaluation on bias-conflicting samples:  Table 4 demonstrates the evaluation results on bias-conflicting samples. This also clears my concerns.
> >
> > 4) Minor issue: The authors provide satisfactory feedback about our suggestion and feedback. Also, the visualization results on BAR dataset show that the proposed algorithm has advantages in real-world dataset.
> >
> > Overall, we initially liked the interesting idea but somewhat penalized on the experimental setting and evaluations. After the rebuttal, the issues on experiments have been resolved, thus we can increase the rating to 7 (Good paper, accept).

---

> > ### Public Comment · ~Yash_Vardhan_Sharma1 · 2021-11-26
> > **Results on CMNIST**
> >
> > Still there is a large discrepancy in results of LfF reported in orignal vs yours.
> > Why is it so?
> > Model used is the same

---

> > > ### Public Comment · Authors · 2021-11-28
> > > **Using a different version of Colored MNIST**
> > >
> > > Thanks for the interest in our work
> > > We used a different Colored MNIST compared to the original paper of LfF
> > > Specific details of the dataset we constructed can be found in our github repository: https://github.com/kakaoenterprise/Learning-Debiased-Disentangled

---

### Official Review · Reviewer_8Ai1 · 2021-07-18

**Rating:** 7
**Confidence:** 4

**Summary:**

The paper proposes a novel feature-level data augmentation technique to train image classifiers in a debiased manner. The approach consists of disentangling “intrinsic” and “bias” attributes from training set images and then synthesizing new “data points” by combining intrinsic and bias features from different images. Since, a lot of image datasets contain a large amount of bias-aligned data points (high correlation between intrinsic and bias attributes), the proposed feature-swapping approach’s aim is to generate bias-conflicting data points. The approach results in better classification performance on 2 synthetic and 2 real-world datasets.

**Ethical Concerns:**

I do not think that there are any ethical concerns related to this work.

**Limitations And Societal Impact:**

Yes, the authors discuss their work’s limitations and societal impact.

**Main Review:**

Strengths:
- The paper tackles an important problem — spurious correlations in image datasets that result in poor generalization in real-world applications.
- The proposed approach is simple yet effective.
- The paper presents promising quantitative results for 4 different datasets, supporting the robustness of their approach.


Weaknesses:
- I think the paper needs much more explanation in lines 159-160 about what GCE is and how it enables learning bias attributes as compared to intrinsic attributes? Similarly, I think the paper should additionally discuss how cross-entropy loss enables learning intrinsic attributes? If CE is enough to learn intrinsic attributes, any typical image classification systems trained with CE should be able to learn disentangled representations and there wouldn’t be any need for debasing?
- Figure 3: Could the authors please include the visualization of z_b for BAR? Since the target labels are available, its coloring with target labels would still make sense. Also, I disagree that this figure (in its current form) shows that the proposed approach successfully disentangles the intrinsic and bias attributes for BAR dataset (referring to line 269).
- Figure 4: are these examples random or cherry-picked? Could the authors please include a randomly chosen 10x10 matrix?
- How are the ratio of bias-conflicting samples determined for each dataset?
- Lines 228: Both encoders and classifiers are MLPs?
- Lines 50-51: The thickness or the degree of tilt are mentioned as intrinsic attributes. How is that? Shouldn’t they be bias attributes? They should not be contributing to the “digit” classes. This raises a bigger question — how should intrinsic and bias attributes be determined for each dataset? By humans? I understand that the proposed approach doesn’t require it but its qualitative evaluation does require it.
- Section 3 shows that 1) larger diversity ratio is better and that 2) larger sampling ratio is better. Why is there a need to compare these two very different things with each other (lines 129-131)?

Overall, although the experiment section shows promising quantitative results, I believe that the paper lacks enough explanation of how their approach works. Therefore, I am a bit hesitant in recommending the paper for acceptance.


-----------------------------------------------------------------------------------------------------------------------------------------------------------------------------------------
-----------------------------------------------------------------------------------------------------------------------------------------------------------------------------------------
I thank the authors for their response. It answers all my concerns/questions. Therefore, I am increasing my rating to accept.

**Time Spent Reviewing:**

4

---

> ### Author Response · Authors · 2021-08-10
> **Response to reviewer 8Ai1 (R1)**
>
> **GCE loss and relative difficulty score W**
>
> The Generalized Cross Entropy (GCE) loss [1] is described as
>
> $GCE(p(x), y) = \frac{1-p(x)^{q}}{q}$,
>
> where $y$ refers to the ground truth label, $p(x)$ indicates the softmax output of the neural network for an image $x$, and $q\in(0,1)$ is a hyper-parameter.
> Compared to the CE loss, the GCE loss imposes high weights on the gradients for the samples which have high probability of the target class $y$.
> Therefore, when training models in a biased dataset, GCE loss emphasizes the training on the easy samples (i.e., bias-aligned samples) with the high probability values for the target label, leading the network to be fully biased.
> In this respect, we train the ($E_{b}$ and $C_{b}$) with the GCE loss, and ($E_{i}$ and $C_{i}$) with the CE loss.
> As ($E_{b}$ and $C_{b}$) are trained to be fully biased, we can obtain a high CE loss when the bias-conflicting samples are given to the ($E_{b}$ and $C_{b}$), while having relatively a small CE loss with the ($E_{i}$ and $C_{i}$).
> Utilizing the Eq. (1) in the main paper, we can obtain the high relative difficulty score $W$ for the bias-conflicting samples.
>
> $W$ is then multiplied with the CE loss obtained from the ($E_{i}$ and $C_{i}$) in Eqs. (2) and (3).
> The main paper omits this multiplication of $W$, so we will revise it in our camera ready.
> The revised Eqs. (2) and (3) are as follows:
>
> $L_\text{dis} = W(x)\cdot CE(C_i(z), y) + \lambda_\text{dis} GCE(C_b(z), y)$,
>
> $L_\text{swap} = W(x)\cdot CE(C_i(z_\text{swap}), y) + \lambda_{\text{swap}_b} GCE(C_b(z_\text{swap}), \tilde{y})$.
>
> According to the revised Eqs. (2) and (3), the CE loss re-weighted with the $W$ can emphasize the training on the bias-conflicting samples and thus enables the learning of intrinsic attributes.
> We want to clarify that using the CE loss without the multiplication of $W$ does not enable to learn the intrinsic attributes since the bias-conflicting samples would not be emphasized during the training.
> We will add such details after lines 159-160 as R1 pointed out.
>
> **Visualization of $z_b$ for BAR**
>
> We provide the 2D projection results of using the bias feature vectors $z_b$ for BAR in REB-FIG1.
> As the ground-truth bias labels do not exist, we first ran the $K$-means clustering on the $z_b$ vectors and obtained their cluster assignments. Then, we colorized them by utilizing such cluster assignments as the bias labels, as shown in REB-FIG1.
> Afterwards, to verify that the data samples are grouped according to the bias factor, we randomly sampled the images from each cluster and provide them in REB-FIG2.
> We found that the images from each cluster share the similar backgrounds/colors, even with different target labels (i.e., action).
> For example, the six images in the upper left corner of REB-FIG2 have different target labels, but they all contain the grass or green background. Similarly, other sets of images are also clustered according to the backgrounds (e.g., ice/white, dirt/brown, and indoor/dark). Through the additional qualitative analysis, we observe that $z_b$ mainly contains the bias attributes, demonstrating that our method disentangles the intrinsic and bias attributes.
>
> REB-FIG1 (2D projection of $z_b$ for BAR): https://tinyurl.com/4037-REBFIG1
>
> REB-FIG2 (Representative images of each cluster): https://tinyurl.com/4037-REBFIG2
>
> **Randomly chosen $10\times10$ matrix of Colored MNIST**
>
> REB-FIG3 shows a randomly chosen $10\times10$ matrix in addition to Fig. (4) of the main paper.
>
> The first column indicates the images used for extracting the intrinsic attribute (i.e., digit), and the first row contains the images used for extracting the bias attribute (i.e., color). Similar to Fig. (4) of the main paper, for a given row, the reconstructed image changes its color while maintaining its digit shape.
> For the purpose of visualization, we conducted a post-hoc training of the decoder, which learns to take the disentangled latent vectors from the pre-trained encoders and reconstruct the images.
> The reconstruction loss only updates the parameters of the decoder, not the ones of the classification network.
> Due to this fact, there may exist reconstructed images with unrecognizable digit shapes or blurry artifacts.
> Yet, REB-FIG3 shows that the reconstructed images reasonably change its color while maintaining its digit shape as we intended.
>
> REB-FIG3 ($10\times10$ matrix of Colored MNIST): https://tinyurl.com/4037-REBFIG3
>
> **Deciding ratios of bias-conflicting samples for each dataset**
>
> We followed the evaluation protocols used in the previous studies which tried to address debiasing [2,3,4,5]. Evaluating models on the various ratios (i.e., 0.5%, 1%, 2%, and 5%) demonstrates the generalization ability of debiased models on diverse biased settings.
>
> **Encoders/classifiers in line 228**
>
> We used the multi-layer perceptron (MLP) with three hidden layers for the encoder, and an additional linear layer is used as the linear classifier. We will elaborate on such details of the architectures in the camera ready.
>
> **Thickness/tilted degrees used as examples of intrinsic attributes**
>
> As R1 pointed out, the thickness or the tilted degrees are not the examples of the intrinsic attributes of digits in that changing these attributes do not change the class of digits in MNIST.
> Appropriate examples of the intrinsic attributes we originally intended would be the spatial structures or shapes of digits. We deeply thank R1 for pointing out such misleading examples and we will revise them in our camera ready.
>
> **Choosing intrinsic/bias attributes**
>
> By its definition, the intrinsic attributes are determined by the target labels in the dataset. Regarding the bias attributes, our method does not require the prior knowledge of them in the training. However, as the reviewer pointed out, existing methods [1,2,3,4] including ours still require the bias attributes in the datasets to be determined by humans for **evaluation**.
> To be specific, an unbiased or a bias-conflicting test dataset can only be constructed when the bias attributes are determined in advance by humans.
> We acknowledge that it remains challenging to 1) determine whether the bias exists and 2) identify the bias type in the dataset without human involvement.
> Still, we believe that our main contribution comes from achieving the state-of-the-art debiasing performance against the existing baselines, even without requiring the prior knowledge on bias during training.
> We will discuss such limitation of this field in the camera ready.
>
> **Comparing diversity/sampling ratio**
>
> The primary reason for comparing the large oversampling ratio with the small diversity ratio and vice versa is to emphasize the importance of diversity of the bias-conflicting samples in debiasing, which was under explored.
> More specifically, increasing the diversity of the bias-conflicting samples can be more crucial in learning debiased representation compared to oversampling these images.
> Oversampling the bias-conflicting images can be regarded as a simplified version of LfF [1] in that oversampling and LfF both emphasize the bias-conflicting samples during training.
> To help understanding our motivation, we additionally report the classification accuracy of the diversity ratio 1\% and the sampling ratio 1\% in Table (1), where none of the debiasing strategies are applied.
>
> From this baseline setting, it is observed that increasing the diversity ratio from 1% to 5% improves the accuracy by 28.06% and 12.68% for Colored MNIST and Corrupted CIFAR10, respectively, which surpasses the improved accuracies of 17.28% and 9.1% by increasing the sampling ratio.
> Bold and italic numbers indicate the best and second-best accuracy, respectively.
>
> #### **Table 1**
>
> #### **[Colored MNIST]**
>
> | Diversity ratio | Sampling ratio | Accuracy |
> | :---: | :---: |:---: |
> | 5% | 50% | **83.77**±2.03 |
> | 1% | 50% | 67.19±1.99 |
> | 5% | 1% | *77.97*±6.00 |
> | 1% | 1% | 49.91±4.22 |
>
> #### **[Corrupted CIFAR10]**
>
> | Diversity ratio | Sampling ratio | Accuracy |
> | :---: | :---: |:---: |
> | 5% | 50% | **46.99**±0.82 |
> | 1% | 50% | 33.08±0.80 |
> | 5% | 1% | *36.66*±0.55 |
> | 1% | 1% | 23.98±0.00 |
>
>
> <References>
>
> [1] [Z. Zhang and M. Sabuncu. Generalized cross entropy loss for training deep neural networks with noisy labels. In Advances in Neural Information Processing Systems, 2018.](https://papers.nips.cc/paper/2018/file/f2925f97bc13ad2852a7a551802feea0-Paper.pdf)
>
> [2] [Junhyun Nam, Hyuntak Cha, Sungsoo Ahn, Jaeho Lee, and Jinwoo Shin. Learning from failure: Training debiased classifier from biased classifier. In Advances in Neural Information Processing Systems, 2020.](https://proceedings.neurips.cc/paper/2020/file/eddc3427c5d77843c2253f1e799fe933-Paper.pdf)
>
> [3] [Hyojin Bahng, Sanghyuk Chun, Sangdoo Yun, Jaegul Choo, and Seong Joon Oh. Learning de-biased representations with biased representations. In International Conference on Machine Learning (ICML), 2020.](https://arxiv.org/pdf/1910.02806.pdf)
>
> [4] [Luke Darlow, Stanisław Jastrze ̨bski, and Amos Storkey. Latent adversarial debiasing: Mitigating collider bias in deep neural networks.](https://arxiv.org/pdf/2011.11486.pdf)
>
> [5] [Enzo Tartaglione, Carlo Alberto Barbano, and Marco Grangetto. EnD: Entangling and Disentangling deep representations for bias correction. In Proceedings of the IEEE/CVF Conference on Computer Vision and Pattern Recognition (CVPR), 2021.](https://openaccess.thecvf.com/content/CVPR2021/papers/Tartaglione_EnD_Entangling_and_Disentangling_Deep_Representations_for_Bias_Correction_CVPR_2021_paper.pdf)
>
> [6] [Haohan Wang, Zexue He, Zachary L. Lipton, and Eric P. Xing. Learning robust representations by projecting superficial statistics out. In International Conference on Learning Representations, 2019.](https://openreview.net/pdf?id=rJEjjoR9K7)

---

### Author Response · Authors · 2021-08-10
**Response to all reviewers**

We deeply appreciate the reviewers for the thoughtful comments and feedback. R1, R2, R3, and R4 indicate Reviewer 8Ai1, Reviewer iKKw, Reviewer ub8A, and Reviewer 3RPm, respectively. Reviewers acknowledged the importance of the problem this paper attempts to address (R1, R4) and found our proposed method “simple yet effective” (R1, R2, R3). Especially, reviewers noted that this paper has potential for high practical impact (R3) and will attract much interest of the NeurIPS community (R4). We discuss how we will reflect the valuable comments in each reviewer-specific response. Additional qualitative results requested are provided as the URL address to the corresponding response of each reviewer. The name of each figure starts with REB-FIG which stands for "rebuttal figures", named in order to differentiate with the figures in the main paper.

The summary of our response is as follows:

**GCE loss and relative difficulty score W (R1, R2, R4)**

Our submission mistakenly omitted the multiplication of W on the CE loss in Eqs. (2) and (3). We revised the equations in each response with additional explanations.

**Quantitative evaluation (R2, R3, R4)**

We conducted additional extensive experiments of 1) addressing the reproducibility of LfF [1], 2) including a new baseline model 'EnD' [2], 3) the evaluation on the bias-conflicting test sets, 4) the evaluation on multiple biases, and 5) the sensitivity of the hyper-parameters. We conducted three independent trials for all the quantitative experiments and reported their averaged accuracies with the standard deviations.

**Additional qualitative analysis (R1, R2, R3)**

We added the 2D projection of $z_b$ on BAR, a $10\times10$ matrix of reconstructed images on Colored MNIST, and the Grad-CAM results on BAR.

Again, we deeply appreciate your comments and we will include the contents in the camera ready. Thanks for the fruitful feedback and improving the quality of our work.

[1] [Junhyun Nam, Hyuntak Cha, Sungsoo Ahn, Jaeho Lee, and Jinwoo Shin. Learning from failure: Training debiased classifier from biased classifier. In Advances in Neural Information Processing Systems, 2020.](https://proceedings.neurips.cc/paper/2020/file/eddc3427c5d77843c2253f1e799fe933-Paper.pdf)

[2] [Enzo Tartaglione, Carlo Alberto Barbano, and Marco Grangetto. EnD: Entangling and Disentangling deep representations for bias correction. In Proceedings of the IEEE/CVF Conference on Computer Vision and Pattern Recognition (CVPR), 2021.](https://openaccess.thecvf.com/content/CVPR2021/papers/Tartaglione_EnD_Entangling_and_Disentangling_Deep_Representations_for_Bias_Correction_CVPR_2021_paper.pdf)

---

### Decision · Program_Chairs · 2021-09-27

**Decision:**

Accept (Oral)

**Comment:**

All reviewers are in agreement that the paper tackles an important problem in a simple yet effective, and enthusiastically recommend acceptance. The authors provided extensive experiments and explanations in their rebuttal which should be incorporated into the final version.